# Blocked training facilitates learning of multiple schemas

Andre O. Beukers[1], Silvy H. P. Collin[2], Ross P. Kempner[1], Nicholas T. Franklin[3], Samuel J. Gershman [3] &
Kenneth A. Norman [1] ✉

We all possess a mental library of schemas that specify how different types of events unfold. How are
these schemas acquired? A key challenge is that learning a new schema can catastrophically interfere
with old knowledge. One solution to this dilemma is to use interleaved training to learn a single
representation that accommodates all schemas. However, another class of models posits that
catastrophic interference can be avoided by splitting off new representations when large prediction
errors occur. A key differentiating prediction is that, according to splitting models, catastrophic
interference can be prevented even under blocked training curricula. We conducted a series of semi-
naturalistic experiments and simulations with Bayesian and neural network models to compare the
predictions made by the "splitting" versus "non-splitting" hypotheses of schema learning. We found
better performance in blocked compared to interleaved curricula, and explain these results using a
Bayesian model that incorporates representational splitting in response to large prediction errors. In a
follow-up experiment, we validated the model prediction that inserting blocked training early in
learning leads to better learning performance than inserting blocked training later in learning. Our
results suggest that different learning environments (i.e., curricula) play an important role in shaping
schema composition.

Over the course of a lifetime, we acquire schematic knowledge of how different events typically unfold; for example, we know what to expect when we go through airport security or order at a restaurant[1–3]. For the purpose of this paper, we define a schema as a learned mental model used for predicting upcoming states in the environment (e.g., ref. [2,4,5]). Mental schemas are powerful devices for remembering the past, interpreting the present, and predicting the future. However, they are hard to learn for several reasons. First, schema learning is (for the most part) unsupervised. We are not explicitly told during learning that we are witnessing an instance of a particular type of event (or even how many such types of events there are); rather, we have to learn to "carve the world at its joints" on our own. Second, sensory features are aliased across schemas: The same perceptual features can have different predictive consequences in different schemas; for example, the etiquette for how to appropriately respond to a ringing phone is different depending on whether you are with friends or in a business meeting.

This aliasing problem leads to the risk of catastrophic interference (CI), the overwriting of old knowledge caused by new learning[6,7]. When knowledge is represented as distributed patterns of weights, different items compete for the same representational resources. If two schemas share features but have different sequential structure, and they are represented in the same set of weights, then learning about the second schema can catastrophically overwrite knowledge about the first schema. One way to avoid CI is to interleave new experiences with old knowledge during learning[8]. By continually pressuring the network weights to maintain old knowledge while encoding new information, interleaved learning allows the weights to settle into a state that jointly represents new and old information. In keeping with these simulation results, numerous studies have shown benefits of interleaved learning (e.g. ref. [9–11]). However, interleaving has been shown to slow down learning (e.g., ref. [8]) and might be infeasible at scale, as the size of the set of experiences that need to be interleaved grows with the number of schemas that are learned (but see[12]).

While interleaved learning prevents CI by accommodating multiple schemas on a shared representational substrate (e.g., set of weights), a different strategy is to represent different schemas using separate representational resources when large prediction errors occur[13,14]. For example, say you travel to a new country where cars drive on the opposite side of the road.

[1]Department of Psychology and Princeton Neuroscience Institute, Princeton University, Princeton, NJ, USA. [2]Tilburg School of Humanities and Digital Sciences, Tilburg University, Tilburg, The Netherlands. [3]Department of Psychology and Center for Brain Science, Harvard University, Cambridge, MA, USA.
✉e-mail: knorman@princeton.edu

This should result in prediction errors based on your existing "driving" schema. In this scenario, you could either update your "driving" schema (which could damage your understanding of how "driving" works) or you could learn about "driving" in this new country using a different set of weights. This latter strategy prevents new learning from interfering with how you already drive, at the potential cost of impeding generalization between "driving" in this new country and "driving" at home. In short, representational overlap affords generalization but risks interference, while splitting reduces interference risk while giving up on representing generalizations.

Several cognitive models have been developed that have this property of "splitting" in response to large prediction errors (Adaptive Resonance Theory[13]; SUSTAIN[15]; Event Segmentation Theory[16]; and Latent Cause Inference[17,18]). One recently-developed model of this sort is the Structured Event Memory (SEM) model[19]. SEM builds up a library of recurrent neural networks (RNNs), where each RNN is specialized for modeling a particular kind of event (i.e., each RNN implements a particular schema). When a new perceptual input is fed into the model, SEM uses Bayesian inference to determine which RNN (schema) is most relevant; SEM then uses that RNN to make predictions and updates the weights in that RNN in response to new inputs. When the currently-selected RNN starts making large prediction errors, SEM initiates a search process for a new RNN (from the "library") that does a better job of predicting. If no suitable RNN is available the model splits off a new RNN to model the new input. In this way, SEM gradually grows its library of schemas to accommodate new kinds of situations.

A key prediction of splitting models is that interleaving is not necessary to avoid catastrophic interference: If the model is given a block of training on inputs that follow one structure, and then switches to training on inputs that follow a new structure, this unsignaled transition should trigger a large prediction error. That, in turn, will cause the model to split off the representation of the new schema from the old schema, preventing CI. By contrast, models without the capacity for splitting predict that CI will happen in this blocked-learning situation – i.e. during the second block, the schema learned in the first block will be overwritten. Note that prior studies that have manipulated blocked vs. interleaved learning have not examined learning of multiple schemas (where schema transitions are not marked and the

features of the schemas are unknown), so this prediction remains to be tested. Here, we set out to test this prediction using a task in which participants viewed narratives that were generated according to two distinct underlying schemas; we manipulated whether stories generated by the two schemas were presented in a blocked or interleaved fashion. If splitting models are correct, participants should perform well in the blocked condition instead of showing catastrophic interference.

## Results
### Overview of schema learning task and results

To operationalize schema learning, we took inspiration from artificial grammar learning (AGL; ref. [20,21]). In AGL, participants learn through repeated exposure to the generative structure of stimulus sequences. Importantly, these stimulus sequences are generated by draws from an underlying Markov chain that operationally defines the generative structure of the environment; the states of the chain determine the observable state of the environment, and the edges of the chain encode the transition structure of the environment. While AGL traditionally uses Markov chains to generate meaningless stimulus sequences (e.g., arbitrary sequence of letters), here we used Markov chains to algorithmically generate narratives (either text narratives or computer-animated movies).

In all of our experiments and simulations, we used two chains (chain A and chain B) with the following structure (Fig. 1): The second time step of each chain uniquely identifies the following sequence as being generated from chain A or B – state 1 only occurs in chain A and state 2 only occurs in chain B. State 0 and states 3-9 occur in both chains, and the observation generated by each of these states is the same regardless of whether that state occurs as part of chain A or B (i.e., these states are aliased across chains; see "story chains" in Methods). Finally, while states are aliased across chains, the transition structure is mirror opposite. For example, while in chain A state 3 transitions to state 5, in chain B state 3 transitions to state 6 (see Fig. 1 for graphical depiction). Therefore, predicting transitions requires a combination of the observable state and also knowledge of which chain is generating the current story (which, in turn, requires memory for previous states in the story). A key feature of the generative model is that the transition from state 1 or 2 to state 3 or 4 is unpredictable (i.e., each possible

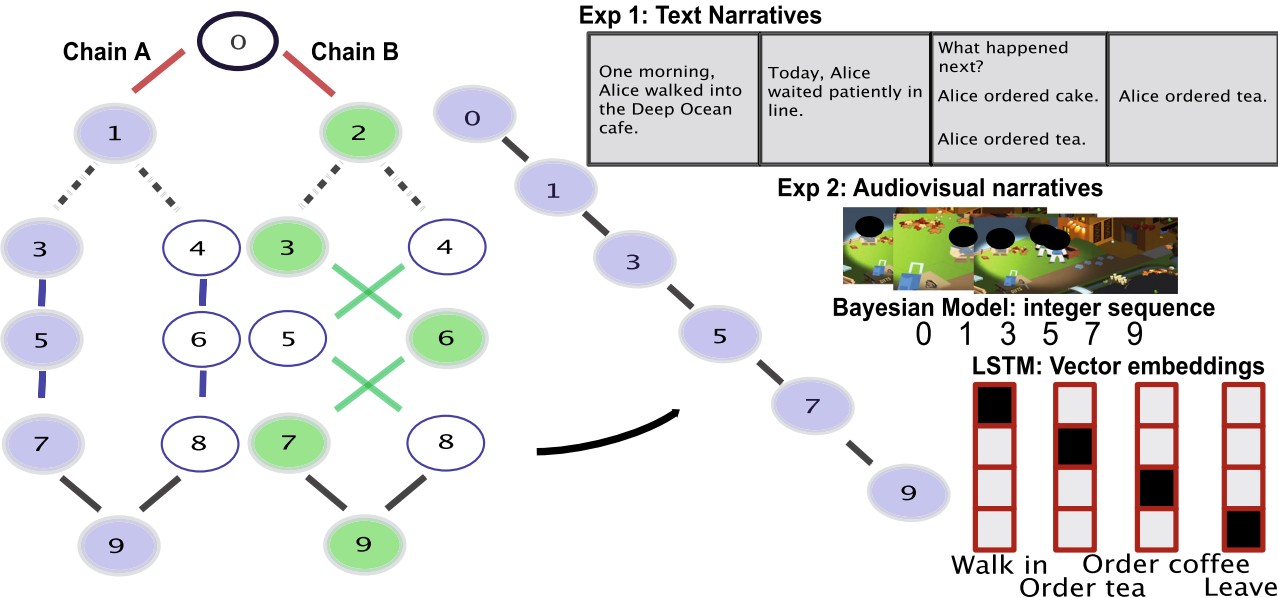

**Fig. 1 | Common task structure used across all experiments and simulations.** Stories were generated from one of two possible structures (chain A and chain B) defined as Markov chains where states are events and edges define transitions between events. For both chains, the transition into states 3 and 4 was probabilistic, while every other transition (up until the very last state 9) was fully determined by the chain. This ensures that the marginal probability of visiting states 3-9 is equal across chains. A draw from one of the chains (curved arrow, blue path) is rendered and presented to participants as a text (exp 1) or audiovisual (exp 2) narratives, and to models as integer sequences or one-hot vector embeddings. For copyright reasons, faces have been covered by black ovals in the audiovisual narrative screenshots shown here.

transition occurs 50% of the time) – this has the consequence that states 3 through 9 are observed equally often in chain A and chain B, so seeing any of these states on their own does not provide any information about which chain is generating the story.

The main manipulation used across all experiments and simulations was the order in which stories were generated using chain A or B (i.e., the curriculum). In blocked curricula, multiple instances from one chain are generated before switching over to the other chain (i.e. A … A B … B). In interleaved curricula, the stories are generated from the two chains in strict alternation (i.e. A B A B). We used this strictly alternating structure to ensure that we did not inadvertently obtain a long run of stories from one chain during training (as this would make learning dynamics more similar across the blocked and interleaved conditions, reducing differences between these conditions).

We predicted that, in blocked curricula, non-splitting models would suffer CI while splitting models might not. We begin by confirming this intuition by simulating a non-splitting model for sequence learning on our task. We then report a behavioral study that finds, in contrast to the predictions made by non-splitting models, that schema learning in human participants is much better under blocked compared to interleaved curricula. We provide multiple exact and conceptual replications of this phenomenon. We formalize a theory of these observations implemented as a non-parametric Bayesian model that incorporates splitting. Our model explains good performance in the blocked condition in terms of splitting occurring in response to large prediction errors at block boundaries (thereby properly "carving nature at its joints"). Notably, our model accounts for poor performance in the interleaved condition in terms of failure to split. In the interleaved condition, the model does not reliably experience large prediction errors when the generative model shifts, so the model often (but not always) fails to split, instead ending up with a single schema that does not align properly with the generative structure of the environment (and thus does not support accurate prediction). Our model also makes the prediction that, when the training curriculum incorporates both blocked and interleaved phases, the order of these phases should matter: Doing interleaved learning before blocked learning should lead to lower asymptotic levels of learning because the initial interleaved learning phase causes the model to form poor schemas, which it then brings to the ensuing blocked phase. This prediction was confirmed, providing further support for the splitting framework.

## Long-short term memory model simulations

In this first simulation, we demonstrate catastrophic interference (CI) after blocked learning in non-splitting models. To do this, we trained a form of

recurrent neural network called a Long-Short Term Memory network (LSTM[22]) to perform the task described in the previous section. Critically, the LSTM architecture has no representational splitting mechanism. As such, when trained on two tasks, it attempts to accommodate the knowledge representation of both tasks within the same representational substrate.

The states of the task chain (Fig. 1) were encoded as one-hot vectors, with all but the first state of each chain being aliased between chains (for a total of 9 entries, each corresponding to a different chain state). On each step of processing, the network is provided with one of these vector representations of a chain state. With each observation, the network first updates its internal state, and then produces an output. The output produced by the network comes as a softmax activation over 9 units (each corresponding to a different chain state), which can be interpreted as a probability distribution over what the network believes will be the next possible state.

The main manipulation was the training curriculum. One group of networks ($N = 10$ random seeds) was trained on a blocked curriculum: 40 draws were taken from chain A, then 40 draws from chain B, then another 40 from A and B respectively, for a total of $4 \times 40 = 160$ stories. A second group of networks ($N = 10$ same random seeds) were trained on an interleaved curriculum, such that the chain being drawn from swapped for every sequence (ABAB…) for 160 consecutive stories. To make for a fair comparison across conditions, the final 40 stories of both conditions were generated according to a random curriculum, in which each story is generated with an equal probability from chain A or B.

To assess CI, we froze the network weights after each story and inspected how the network would respond to inputs from both chains. Figure 2 plots the network's softmax activations, interpreted as the network's probability estimate that the next state will be the state that follows the current one according to chain A vs. chain B, after being given test inputs that were actually generated from chain A (left) or chain B (right) under interleaved (top) or blocked (bottom) training. Under interleaved training, the network eventually learned to respond appropriately to paths generated from both chains. Going into the final test period (corresponding to story 160 on the $x$-axis), the network assigns high probability to the transition corresponding to the chain in which it is being tested. However, when trained under a blocked curriculum, the network's response disregards the test input and instead always corresponds to the chain it was being trained on. Going into the final test period at story 160, the network assigns high probabilities to transitions corresponding to the chain in which it was just trained on (green background indicates training on chain A). Notice that, regardless of whether the network is given an evaluation input corresponding to chain A (left) or B (right), it flipflops and assigns high probability to the transition that corresponds to the chain it is currently being

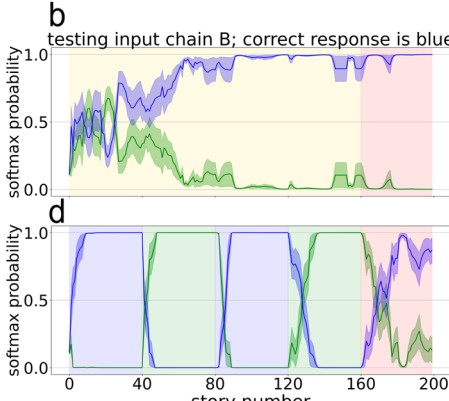

**Fig. 2 | LSTM simulations showing catastrophic interference during blocked training in a non-splitting model.** LSTM networks were trained on interleaved (**a**, **b**) and blocked (**c**, **d**) curricula. While each training epoch was conducted on an output generated by a single chain, networks were continually evaluated throughout training on sequences from chain A (**a**, **c**) and from chain B (**b**, **d**). Background color indicates training curriculum: Yellow is perfectly interleaved training, green is training on chain A only, blue is training on chain B only, and red is the random-curriculum test phase (50% chance of chain A, 50% chance of chain B). X-axis is training epoch, corresponding to the presentation of a single sequence, $y$-axis is the softmax activation, interpreted as the network's probability estimate that the next state will be the "chain A" (green line) or "chain B" (blue line) state that follows the current state. Error ribbons indicate ± 1 standard error.

trained on (indicated by the background color). This pattern of results is consistent with the interpretation that blocked training causes knowledge of chain A and knowledge of chain B to interfere. It is also interesting to note that, during the test phase (stories 161–200), because the network is seeing samples from chain A and chain B close together in time, it can begin to learn to distinguish these two – this is evidenced by the higher probability assigned to the chain it is being tested on by story 200.

## Text narrative experiments

The previous simulations confirm the prediction that non-splitting models suffer from CI during blocked training, while interleaved training should lead to near-ceiling performance. We tested this prediction in a behavioral task conducted with participants on Amazon Mechanical Turk (AMT). While the above simulations embedded the chain states as one-hot vectors, here those same states were used to generate sentences that together composed a story. For these experiments, the identity of the schema (chain) was signified by the location, which is revealed at the start of the story: State 1 (which is only seen in chain A) produces the text observation "[subject] walked into the Jungle Brew House", while state 2 (which is only seen in chain B) produces "[subject] walked into the Deep Ocean Cafe" (note that the identity of the "subject" was set to a unique value for each story; see Methods). We did not inform participants in this study that the location signified schema identity, nor did we inform them of the number of underlying schemas or the structure of the schemas – participants had to learn all of this on their own.

Participants read 200 such stories, one sentence at a time, and continually made 2-alternative forced choice (2AFC) predictions about what will happen next in the story (see Fig. 3 for numbers of participants in each condition). The main manipulation was again training curriculum: In the interleaved condition, the chain generating stories alternated every story; in the blocked condition, story presentation was blocked by chain, so that chains alternated every 40 stories. After 160 stories that were delivered according to either the interleaved or blocked curriculum (the "training" period), the final 40 stories in both conditions (the "test" period) were generated by randomly choosing a chain with 50% probability for each story. Importantly, the transition from the "training" period to the "test" period was not signaled in any way to participants.

Results from the study are shown in Fig. 3 (since states 3 and 4 were not predictable, the figure only reports prediction accuracy for states 5, 6, 7, and 8). We found that participants had much higher accuracy in the blocked compared to the interleaved condition. Importantly, in the random-curriculum test phase, which was matched between conditions (and where the environment is switching at a frequency much more similar to the interleaved condition), prediction accuracy was higher in the blocked training group ($M = 0.884$, 95% CI = [0.826, 0.943]) compared to the interleaved training group ($M = 0.593$, 95% CI = [0.546, 0.641]; t(70) = 7.64, $p < 0.001$, $d = 1.81$, 95% CI = [0.215, 0.367]). Note however that the interleaved group was above chance (t(38) = 3.82, $p < 0.001$, $d = 0.612$, 95% CI = [0.0439, 0.143]). We ran an exact, preregistered replication (see Methods for link to preregistration) and found the same pattern of results: Blocked training ($M = 0.901$, 95% CI = [0.848, 0.955]) resulted in better learning than interleaved ($M = 0.646$, 95% CI = [0.569, 0.724]; t(48) = 5.38, $p < 0.001$, $d = 1.52$, 95% CI = [0.160, 0.351]), but interleaved was above chance (t(23) = 3.70, $p = 0.00119$, $d = 0.755$, 95% CI = [0.0644, 0.228]). Supplementary Figs. 1 and 2 show the results separately for predictions of states 5/6 and states 7/8, respectively – both of these show the same patterns as the results shown in Fig. 3.

Note that the results shown in Fig. 3 were for a filtered sample of participants who passed attention checks (see Section "Text Narrative Experiments: Attention Check And Exclusion Criterion"). Importantly, the results were qualitatively the same without any participant exclusions: Blocked learning led to better performance than interleaved learning for both the initial sample (t(103) = 5.87, $p < 0.001$, $d = 1.15$, 95% CI = [0.139, 0.282]) and the replication (t(117) = 4.02, $p < 0.001$, $d = 0.739$, 95% CI = [0.0715, 0.210]), and the interleaved groups in both the initial sample and replication were still above chance (t(54) = 4.32, $p < 0.001$, $d = 0.582$, 95% CI = [0.0460, 0.126]; t(55) = 3.66, $p < 0.001$, $d = 0.488$, 95% CI = [0.0329, 0.113]); see Supplementary Fig. 3.

To rule out the possibility that participants in the interleaved condition are simply not attending to the information in the first state that is diagnostic of chain identity (in this case, story setting), we ran a further interleaved condition in which we explicitly instructed participants to attend to the schema-identifying feature ("pay attention to the story setting"), and included the schema-identifying feature (i.e., the story setting) in bold-face

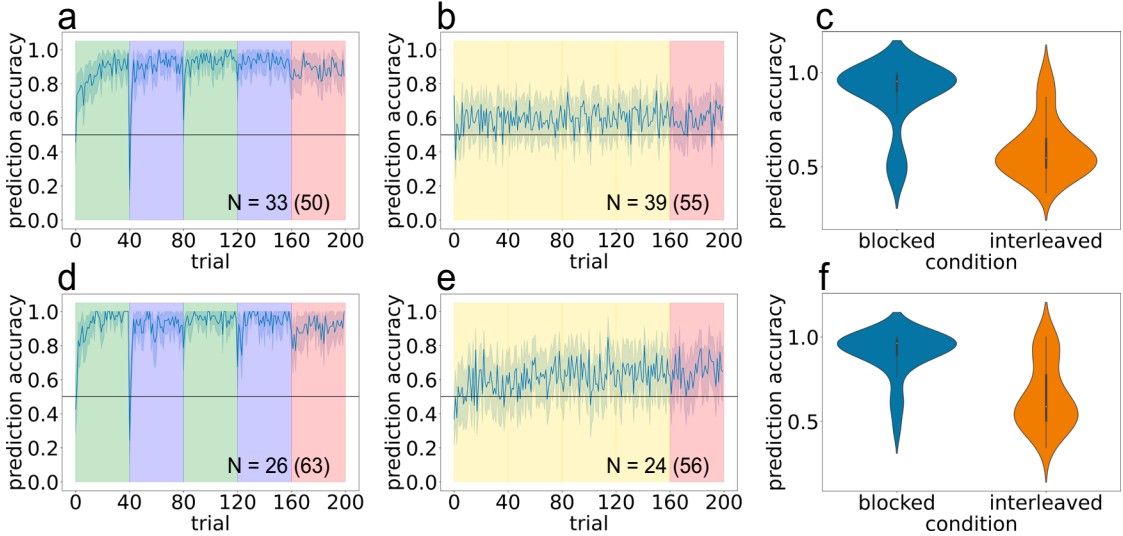

**Fig. 3 | Text narrative experiments show better schema learning in humans trained on blocked compared to interleaved curricula.** Initial results (**a**–**c**) replicated in independent sample of participants (**d**–**f**). Left column (**a**, **d**) shows blocked results; middle column (**b**, **e**) shows interleaved results. **a**, **b**, **d**, **e** Mean accuracy across participants over time. *Y*-axis is between participant average accuracy, *x*-axis is time (200 stories). Background colors indicate training curriculum: green is training on chain A only, blue is training on chain B only, yellow is interleaved training, and red is the random-curriculum test phase (50% chance of chain A, 50% chance of chain B). *N* indicates the number of participants included in the final analysis; the number in parentheses indicates the total number of participants before exclusion (see attention check and exclusion criteria in Methods). Error ribbons indicate ± 1 standard error. **c**, **f** Violin plots showing accuracy distribution during test phase.

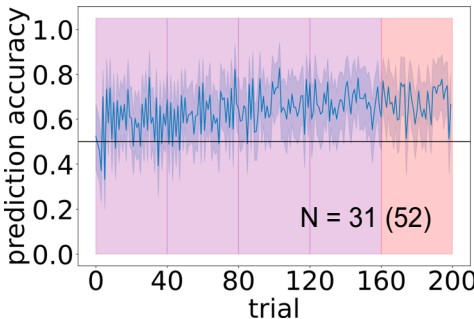

**Fig. 4 | Explicitly instructing participants to attend to the schema-identifying feature does not improve learning on interleaved curriculum.** Purple indicates interleaved curriculum (strictly alternating between A and B) and red indicates the random-curriculum test phase (50% chance of chain A, 50% chance of chain B). Mean accuracy across participants. *Y*-axis is between participant average accuracy, *x*-axis is time (200 stories). N indicates the number of participants included in the final analysis; the number in parentheses indicates the total number of participants before exclusion (see attention check and exclusion criteria in Methods). Error ribbons indicate ± 1 standard error.

at the beginning of every sentence of every story (e.g., "In the cafe, Alice sat down and read a book"). Results are shown in Fig. 4. Performance was significantly above chance (*M* = 0.682, 95% CI = [0.608, 0.755], t(30) = 4.83, *p* < 0.001, *d* = 0.867, 95% CI = [0.105, 0.258]). Crucially, although performance was higher than in the original interleaved condition (t(68) = 2.04, *p* = 0.0454, *d* = 0.490, 95% CI = [0.00185, 0.175]), performance was still significantly lower than in the blocked condition from the original experiment (t(62) = 4.26, *p* < 0.001, *d* = 1.07, 95% CI = [0.108, 0.298]). When we analyzed the data without participant exclusions, explicit interleaved performance was worse than in the blocked condition from the original experiment (t(100) = 4.38, *p* < 0.001, *d* = 0.867, 95% CI = [0.0961, 0.255]), and there was not a statistically significant difference between performance in the explicit interleaved condition and performance in the original interleaved condition (t(105) = 1.07, *p* = 0.287, *d* = 0.207, 95% CI = [−0.0297, 0.0995]). Taken together, these results suggest that worse learning in the interleaved condition (compared to the blocked condition) does not arise from a failure to attend to the schema-identifying feature.

### Experiment 2: animated narrative experiments

To test for the robustness of these results, we ran a conceptual replication. While the previous experiment used text narratives created using the generative model in Fig. 1, here we used the same generative model to create audiovisual narratives about a couple getting married in a fictional island, where each state corresponded to a particular (invented) wedding ritual. Whereas, in the previous experiment, the schema-identifying feature was the location (Deep Ocean Cafe vs Jungle Brew House), here the schema-identifying feature (as before, revealed at the outset of the narrative) is whether the wedding is from a couple from the North or the South island (as before, participants were not informed that North / South determined the schema identity, nor were they informed of the number of underlying schemas or any other properties of the schemas – participants had to learn all of this on their own). Because these audiovisual narratives take longer to present than the text narratives, this study was also compressed: While participants in the previous experiments read 200 stories, participants in this study watched 36 animated narratives (hereafter, movies).

This study was conducted over two days. During the first day, participants viewed 24 movies. In the blocked condition, 12 movies from chain A were followed by 12 movies from chain B. In the interleaved condition, movies from chain A successively alternated with movies from chain B. This first day serves the same purpose as the "training period" (stories 1–160) from the text narrative study, giving participants experience with either a blocked or interleaved curriculum. During the second day, participants in both groups viewed an additional 12 movies, alternating with two movies

from each chain (i.e. A A B B A A...). This second day serves the same purpose as the "test period" (stories 161–200) from the text narrative study. At the end of the viewing session on the second day, participants were asked to respond to forced-choice (FC) questions that probed their knowledge of the transition structure: e.g., "if a couple from the North just celebrated around a campfire, what will most likely happen next?". In the first version of the animated narrative experiment, participants received the two possible events from the next "layer" of the graph as response options and were forced to choose between them. In the second and third versions of the animated narrative experiment, participants were provided with a response set; instead of giving a discrete response, they specified a probability distribution (i.e., distributing 100 "points" across available answer options).

This experiment was conducted on three independent samples. The first two samples were conducted using Amazon Mechanical Turk (AMT) participants, and the third sample was conducted using undergraduate students. Results are shown in Fig. 5. As before, participants in the blocked condition made correct predictions significantly more often compared to the interleaved condition as indicated by significantly better performance on the FC prediction questions (t(46) = 2.14, *p* = 0.0375, *d* = 0.623, 95% CI = [0.0129, 0.413]). These effects were replicated in an independent sample of AMT participants (t(125) = 2.13, *p* = 0.0352, *d* = 0.378, 95% CI = [0.00750, 0.206]) and another independent sample of Princeton undergraduate students (t(28) = 2.90, *p* = 0.00721, *d* = 1.06, 95% CI = [0.0724, 0.421]).

Note that, in the first two experiments, participants were given a 2-alternative forced choice between the possible states that could occur at the next time step (e.g., after observing state 3 or 4, they were asked to choose between states 5 and 6). In the latter two studies, participants were given a 6-alternative response set corresponding to all of the possible states other than the start, end, and schema-identifying states (so, states 3, 4, 5, 6, 7, 8). This allowed us to analyze the error patterns in the interleaved group more closely. The results of this error analysis, shown in Fig. 6, indicate that participants had good knowledge of which two rituals could occur at a given time step, but frequently responded with the ritual associated with the other chain. For example, state 3 was followed by state 5 in chain A (North island) and state 6 in chain B (South island); when participants in the interleaved condition were asked which state followed state 3 when the couple was from the North, they were much more likely to erroneously report state 6 (the state associated with the other chain at that time step) than states 3, 4, 7, and 8 (which occurred at other time steps). The AMT interleaved participant group assigned 0.27 probability to the "wrong chain" state (i.e., the state that occurred at that time step for the other chain); this was well above the probability that they assigned, on average, to each of the "wrong time step" states, t(60) = 7.93, *p* < 0.001, *d* = 1.02, 95% CI = [0.151, 0.253]. Similarly, the undergraduate interleaved participant group assigned 0.39 probability to the "wrong chain" state; this was well above the probability that they assigned, on average, to the each of the "wrong time step" states, t(14) = 7.15, *p* < 0.001, *d* = 1.85, 95% CI = [0.235, 0.437]. The same pattern was present in the blocked condition: The AMT blocked participant group assigned a higher probability to the "wrong chain" state than the probability they assigned, on average, to each of the "wrong time step" states, t(65) = 8.84, *p* < 0.001, *d* = 1.09, 95% CI = [0.170, 0.269]. Similarly, the undergraduate blocked participant group assigned a higher probability to the "wrong chain" state than the probability they assigned, on average, to each of the "wrong time step" states, t(14) = 4.48, *p* < 0.001, *d* = 1.16, 95% CI = [0.114, 0.323].

### Bayesian model simulations: overview

To explain the above results, we implemented a non-parametric Bayesian model (for details, see Methods). Our model learns a library of latent causes; for each latent cause, the model keeps track of the transitions that were observed when that latent cause was active. On each trial, the model is presented with a sequence of states (corresponding to a single story) generated by one of the chains shown in Fig. 1. On each timestep, the model observes a state and attempts to predict the following state. To do this, the model selects the most probable latent cause and uses the empirically

**Fig. 5 | Animated narrative experiments replicate better schema learning in humans trained on blocked compared to interleaved curricula.** Mean accuracy during the test phase for blocked (blue) and interleaved (orange) groups across three independent samples; the dashed line indicates chance performance. *N* indicates the number of participants included in the final analysis (see Methods for information on how exclusions were made).

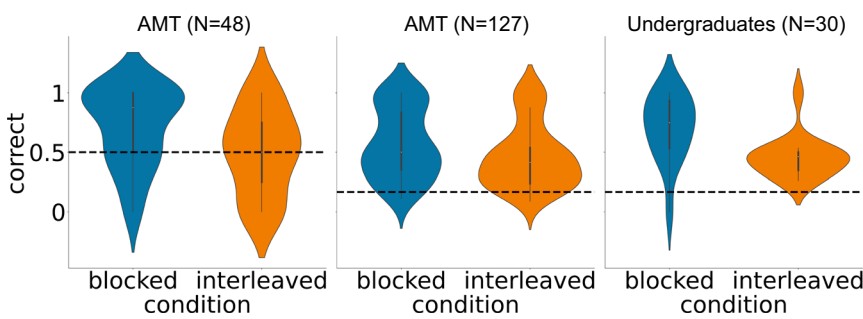

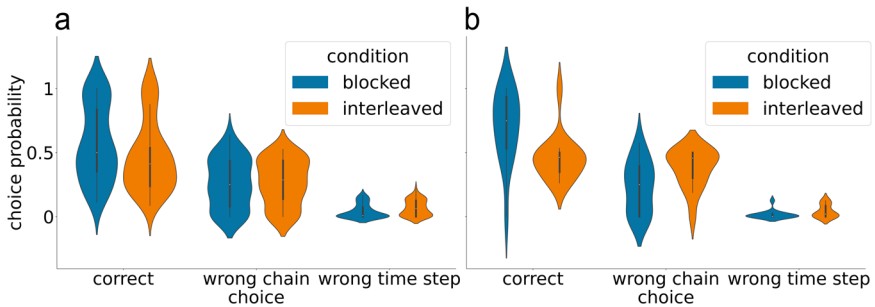

**Fig. 6 | Error analysis for the animated narrative experiments that let participants choose from 6 alternatives. a** shows the choice probabilities for the AMT study; **b** shows the choice probabilities for the study that used undergraduate participants. "Wrong chain" refers to choosing the state associated with the other chain at that time step; "wrong time step" refers to choosing a state that occurs on a different time step. Note that there were four "wrong time step" states but only one "wrong chain" state; to put these results on equal footing, we divided the overall probability of choosing a "wrong time step" state by four -- this gave us the probability of choosing a particular "wrong time step" state.

estimated state transition matrix for that latent cause to calculate the probability of each possible transition. After issuing a prediction and observing the environment transition, the model then updates the posterior probability of each cause in its library, where this posterior probability is a function of the likelihood of the observed transition under each cause and its prior probability; the model also estimates the probability that a new latent cause (not previously used) is present.

Intuitively, the model's prediction error is inversely proportional to the likelihood of the observed transition under the currently-active latent cause. When a large prediction error is observed, that prediction error makes the current latent cause improbable; this, in turn, increases the odds that the most probable latent cause will be a different, previously-used latent cause (in which case the model will switch to that cause) or the new, never-before-used latent cause (in which case the model will split off a new latent cause). Motivated by previous research in latent cause inference, we used a sticky Chinese Restaurant Process (sticky CRP) prior[18,23]. The key features of this prior are that (i) the currently active latent cause is more likely to remain active (i.e. latent causes are sticky), (ii) there are no limits on the number of latent causes that can be inferred, (iii) it possesses a "rich gets richer" property, whereby latent causes that were frequently active in the past have a higher prior probability.

The model has three parameters: $\lambda$ ("sparsity"), $\beta$ ("stickiness") and $\alpha$ ("concentration"). $\lambda$ determines how the transition counts for a new latent cause are initialized and can have a range of values from small (indicating a bias towards extreme transition probabilities) to large (indicating a bias towards less extreme transition probabilities). $\beta$ affects the probability that the latent cause active on the previous timestep/story will remain active. $\alpha$ is used to control the prior probability of splitting off a new (not previously used) latent cause. To fit these parameters, we conducted a parameter search that consisted of sampling a set of parameter values, running the model on 100 simulated experiments to get a mean accuracy, and computing the mean squared error (MSE) between the model's mean accuracy and human participants' mean accuracy at each point in the blocked and interleaved conditions of the original text narrative experiments (looking at both the

training phase and the test phase). Below we report the results obtained using the model parameters that achieved lowest MSE (best fit) against human data. As described below, the initial version of the model did not provide a good overall fit, but we were able to modify the simulation to provide a better account of the data.

**Bayesian model simulation 1: initial challenges with obtaining a good fit**

Our first set of simulations revealed that the model fails to fit the data well in the blocked condition, starting from the second block of training (MSE = 0.0919; see Fig. 7). After having observed the 0–1 transition on the first 40 trials, the first time the model observes the 0–2 transition (at the beginning of the second block on trial 41), it experiences a large prediction error and spawns a new latent cause. The trial goes on and the model observes and records the rest of the transitions observed on that trial (e.g., 0-2-3-6; see Fig. 1) in this newly spawned latent cause. So far this is the expected behavior. However when the model observes a different transition out of the 2 state (e.g., 0-2-4-5) this second sequence has a low likelihood under this new latent cause. Because the CRP prior has a "rich-gets-richer" property, the first latent cause (which has been previously used on the 40 trials in block 1) dominates the posterior. As a consequence, the first latent cause, which encodes the transitions of the first schema, is inadvertently selected. The result of these dynamics is that the model performs well on the first chain, but poorly on the second.

**Bayesian model simulation 2: turning off schema inference at unpredictable transitions**

Thinking back to the structure of the environment, we reasoned that participants might be able to learn that some transitions are unpredictable while others can be learned. Specifically, we hypothesized that – after the first block of seeing that state 1 leads equally often to states 3 and 4 – participants might come to the conclusion that the transition into state 3 and 4 is inherently unpredictable, and (consequently) that they should not factor prediction errors experienced at this transition into their latent cause

inference. To instantiate this, in our second simulation, we switched off the schema inference process at the transition into states 3 and 4 (but then resumed inference for the following time steps). In this simulation, for simplicity, we turned off inference at the transition into 3/4 in both the blocked and interleaved conditions, right from the start of learning; to implement the idea that blocked learning is needed to notice the unpredictability of this transition, we also ran simulations where we only turned off inference for the 3/4 transition after a block of 40 trials from the same chain, and the results were qualitatively similar to the ones presented here (see Supplementary Fig. 4 for results).

After fitting the model parameters, we observed this model variant was better able to fit the average prediction accuracy curve for the blocked condition, in addition to providing a good fit to average human performance in the interleaved condition (MSE = 0.0406; Fig. 8). Looking at the full trajectory of training accuracy across trials, the model succeeds in capturing the large dip in performance at the first block boundary in the blocked condition. There were, however, a few minor discrepancies between the model fits and human performance: The model predicts a somewhat steeper rise in performance in the first block than is apparent in human data. Also, the model slightly underestimates the size of the dip in performance at the first block boundary, and it does not capture the smaller dips in

performance that are evident in the human data at the second and third block boundaries and at the start of the test phase (to facilitate these comparisons, Supplementary Fig. 5 shows the best model fit and the human data overlaid on the same plot). Speculatively, these small discrepancies can be attributed to human participants being inattentive (e.g., failing to notice a shift from the Cafe to the Brew House) whereas the model does not suffer from attentional lapses.

An interesting feature of the model's fit is that it achieves 60% mean accuracy in the interleaved condition via a bimodal distribution, where some model seeds perform close to 100% and a larger number of seeds perform closer to chance (50%; see Fig. 8c) – this bimodality also appears to be present, albeit to a less stark degree, in the human data (see Fig. 3 panels c and f). This observation of bimodality in the interleaved condition both in the model and in the human data raises the question of the *correspondence* between these results – i.e., can the model be used to predict which human participants will succeed or fail in the interleaved condition. Note that the only source of variability from seed to seed in this version of the model is in the stimulus sequence that it experiences (because of the random 3/4 transition) – different seeds experience different transitions from the 1/2 states into the 3/4 states, which then determine (with 100% probability) the states that follow the 3/4 state. Inspection of the model showed that some

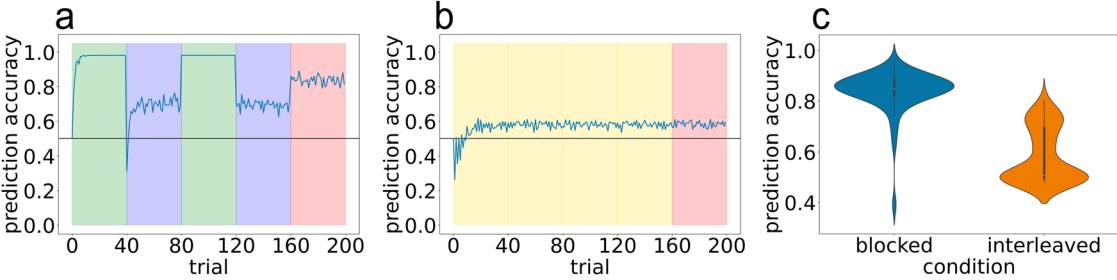

**Fig. 7 | Results from Bayesian model simulation 1 showing poor fit to human data from blocked condition.** Simulation results from best-fitting model parameters on (**a**) blocked and (**b**) interleaved curricula. Background color indicates the training curriculum: Yellow is perfectly interleaved training, green is training on chain A only, blue is training on chain B only, red is the random-curriculum test phase (50%

chance of chain A, 50% chance of chain B). *X*-axis is trial, corresponding to the presentation of a single sequence, *y*-axis is the model's accuracy in predicting the states in the sequence. **c** Violin plot summarizing test accuracy performance of 100 model simulation runs.

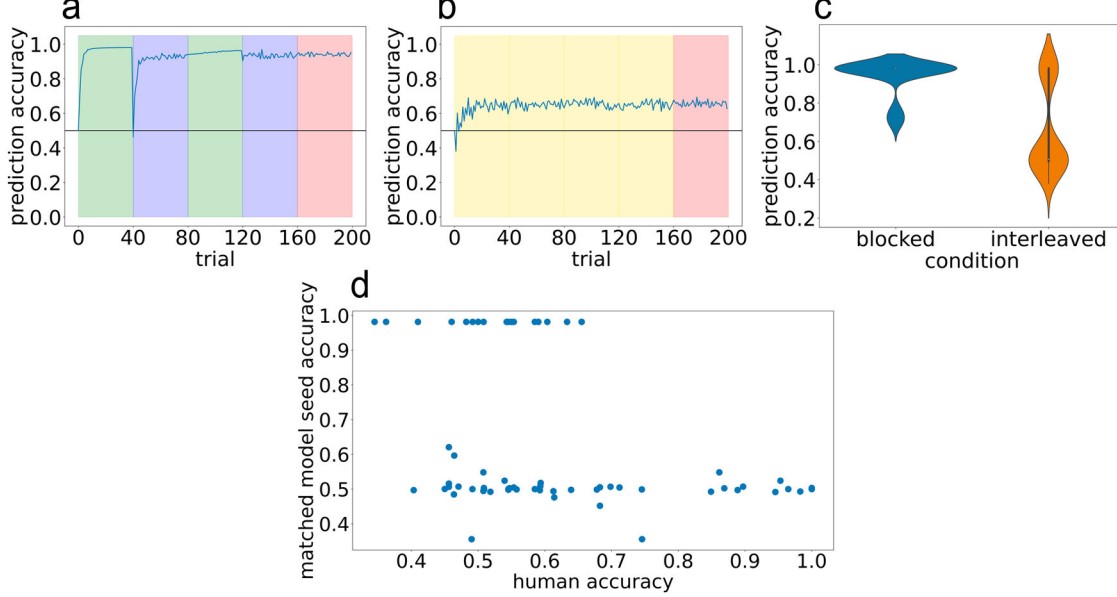

**Fig. 8 | Results from Bayesian model simulation 2.** Removing schema inference on the unpredictable timestep improves model fit to human blocked data. **a, b** Model accuracy over time on blocked and interleaved curricula, respectively. **c** Violin plot

showing model accuracy during test phase. **d** Scatter plot of human accuracy against model accuracy for yoked sequences. Each point corresponds to a pair of human accuracy (*x*-axis) against yoked model accuracy (*y*-axis).

sequences lead to successful learning and other sequences lead to failure. To assess the correspondence between human data and the model, we ran model simulations where – instead of generating new sequences for each model run – we took the exact sequences experienced by our human participants in the original interleaved experiment and gave those to the model. We then compared the accuracy of each human participant to the accuracy of the corresponding model simulation. The predicted relationship did not hold: As shown in Fig. 8d, the sequences that were associated with high accuracy in the model were not reliably associated with higher accuracy in humans. If anything, the relationship was *opposite* to what we predicted, which suggests that sequence variability might play a role in determining human accuracy, but not in the way predicted by this model variant.

### Bayesian model simulation 3: individual differences in model parameters

As described above, Simulation 2 relied entirely on sequence variability to explain variability in human performance; this approach led to incorrect predictions about variability across sequences, so we reject this approach despite it leading to a good fit to the average human performance curve (measured in terms of mean squared error). In Simulation 3, we explored the viability of an alternative approach to modeling human variability, based on individual differences in underlying cognitive parameters. That is, instead of assuming (as we did in the first two simulations) that all people have the same parameters that govern latent cause inference, here we allowed for the possibility that individuals have different values of the concentration parameter (which, as noted above, affects the prior probability of splitting off a new latent cause). To do this, we modified our simulation such that each simulated model had a different concentration value; specifically, for each model run, we sampled a different concentration value centered around a mean value arrived at by our parameter search procedure.

The results from this simulation are shown in Fig. 9. Overall, the model provided a good fit to the mean performance levels in the blocked and interleaved conditions (MSE = 0.0371). As in Simulation 2, the model predicted a bimodal distribution of performance in the interleaved condition, but here – unlike in Simulation 2 – variance in model performance was driven by across-seed variance in the concentration parameter, not by variance in the underlying sequences. We found that models with lower concentration inferred a single latent cause for all environment observations. This "over-lumping" led these models to have a single transition

matrix that simply averaged across transitions. As a consequence, these models had accuracies that remained at chance. In contrast, models with higher concentration values were able to correctly discover two latent causes and score with 100% accuracy. Figure 9 parts d, e, and f illustrates the close correspondence between concentration, number of inferred latent causes, and prediction accuracy. Supplementary Fig. 6 shows a histogram of the number of latent causes inferred in the different conditions of the experiment, confirming that the model mostly (though not always) succeeded in inferring two latent causes in the blocked condition, whereas it was less prone to find this two-cause solution in the interleaved condition; the figure also shows that, when the model did find a two-cause solution, the learned transition matrices matched the ground-truth transition structure, whereas when it found a one-cause solution (in the interleaved condition) the learned transition matrix averaged across the two ground-truth structures (and thus did not accurately approximate either of them).

In other simulations, we found that allowing the stickiness and sparsity parameters to vary across model runs also leads to good overall model fits (Supplementary Fig. 7 shows the results for varying-stickiness simulations, and Supplementary Fig. 8 shows the results for varying-sparsity simulations; quantitatively the MSE was similar across conditions: 0.0376 for varying-sparsity and 0.0377 for varying-stickiness, compared to 0.0371 for varying-concentration). These results show that there is nothing special about the concentration parameter per se – all three of the main model parameters affect whether splitting occurs when the generative model shifts in the interleaved condition, and consequently variance in any of these parameters can affect whether learning succeeds or fails in the interleaved condition.

Overall, the results of these simulations provide a "proof-of-concept" that individual differences in underlying parameters can explain bimodality in performance in the interleaved condition. Importantly, these model results do not show that this explanation is correct – showing this would require us to first derive estimates of the relevant parameters in individual human participants using a separate task, and then relate these parameter estimates to the same participants' performance in our interleaved learning task; this is a topic for future research.

### Bayesian model simulations: interim conclusion

These simulations, taken together, show that a simple, three-parameter Bayesian model incorporating latent cause inference can account for overall patterns of performance in the blocked and interleaved conditions. The

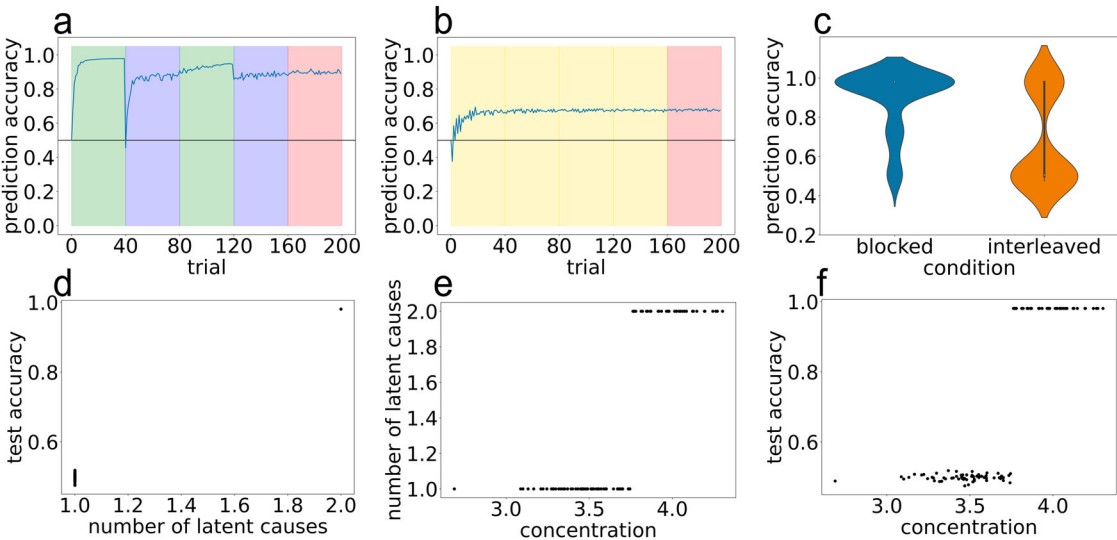

**Fig. 9 | Results from Bayesian model simulation 3.** Individual differences in model concentration parameters can potentially explain human performance in the interleaved condition. **a**, **b** Model accuracy over time on blocked and interleaved curricula, respectively. **c** Violin plot of model test accuracy. **d–f** Increased concentration leads the model to split more regularly in the interleaved condition, which

improves performance. Each dot represents a model run using a slightly different concentration parameter. **d** Number of latent causes versus test accuracy. **e** Concentration versus number of latent causes. **f** Concentration versus test accuracy.

superior overall fits shown in the second and third simulations (which ignored the 3/4 transition when making inferences) compared to the first simulation (which did not ignore the 3/4 transition) suggests that learning to ignore irreducibly unpredictable transitions may be an important determinant of human performance on this task. The simulations also show that we have more work to do before the model provides a compelling account of human variability in performance on these tasks, but they also point to promising directions to explore in the future (e.g., measuring individual differences in latent cause inference and relating this to task performance).

### Inserted blocks simulations
Having demonstrated that Bayesian latent cause models can account for differences in learning between the blocked and interleaved conditions, we next assessed whether the best-fitting model (from Simulation 3) could predict performance given other curricula besides "pure" blocked or interleaved learning. Specifically, we used the model parameters that were shown to best fit the blocked and interleaved conditions above and we ran simulations using mixed curricula. In these curricula, we inserted a pair of blocks (one from each latent cause) at the beginning, middle or end of training. In the "early" condition, the first 40 stories came from the first latent cause, the next 40 stories came from the second latent cause, and these were followed by 80 stories interleaving both latent causes. In the "middle" condition, the first 40 stories were interleaved, the next 40 came from the first latent cause, the following 40 came from the other latent cause, and this was followed by another 40 interleaved stories. In the "late" condition, the first 80 stories were interleaved, followed by 40 from the first latent cause and 40 from the second latent cause. As before, the final 40 test trials were randomly drawn.

Figure 10 panels a–d show the model's predictions for this condition. Most importantly, the model predicts the "early" condition should have the highest accuracy, whereas the "middle" and "late" condition should have lower test accuracies. The "early" condition shows good performance for the

same reason that blocked training shows good performance; exposure to blocks at the start of learning leads the model to properly segment its experience into two latent causes that align with the generative model. The "middle" and "late" conditions show worse performance for the same reason that interleaved training shows worse performance: The initial interleaved training period causes the model to lump together the two schemas into a single latent cause (at least in a subset of simulated participants); crucially, the model is unable to recover from this poor segmentation when it later receives blocked training. The fact that the same (lumped-together) latent cause was used throughout the initial, interleaved part of the training period gives this latent cause a very high prior probability, making it harder for the model to split off a new latent cause when the block boundary finally occurs.

### Inserted-blocks experiments
To test these predictions, we ran an analogous "inserted blocks" experiment on human participants (Fig. 10). As with the simulations, training was divided into four blocks of 40 stories each; in the early condition (A B I I) an initial blocked training was followed by interleaved training, in the middle condition (I A B I) blocked training was inserted half way through interleaved training, and in the late condition (I I A B) blocked training was inserted at the end of interleaved training. As in previous experiments, the final 40 stories for all conditions were generated by a random curriculum, where chain A or B was chosen at random to generate each story.

Confirming the predictions made by our model, participants in the early condition learned better than participants in the middle condition ($t(39) = 3.32$, $p = 0.00197$, $d = 1.04$, 95% CI = [0.0731, 0.301]), and better than participants in the late condition ($t(43) = 2.54$, $p = 0.0150$, $d = 0.758$, 95% CI = [0.0296, 0.260]). This effect replicated in an independent sample (early > middle $t(44) = 5.13$, $p < 0.001$, $d = 1.51$, 95% CI = [0.152, 0.349]; early > late $t(32) = 3.69$, $p < 0.001$, $d = 1.32$, 95% CI = [0.0869, 0.301]). Furthermore, participants in the middle and late condition did not show a

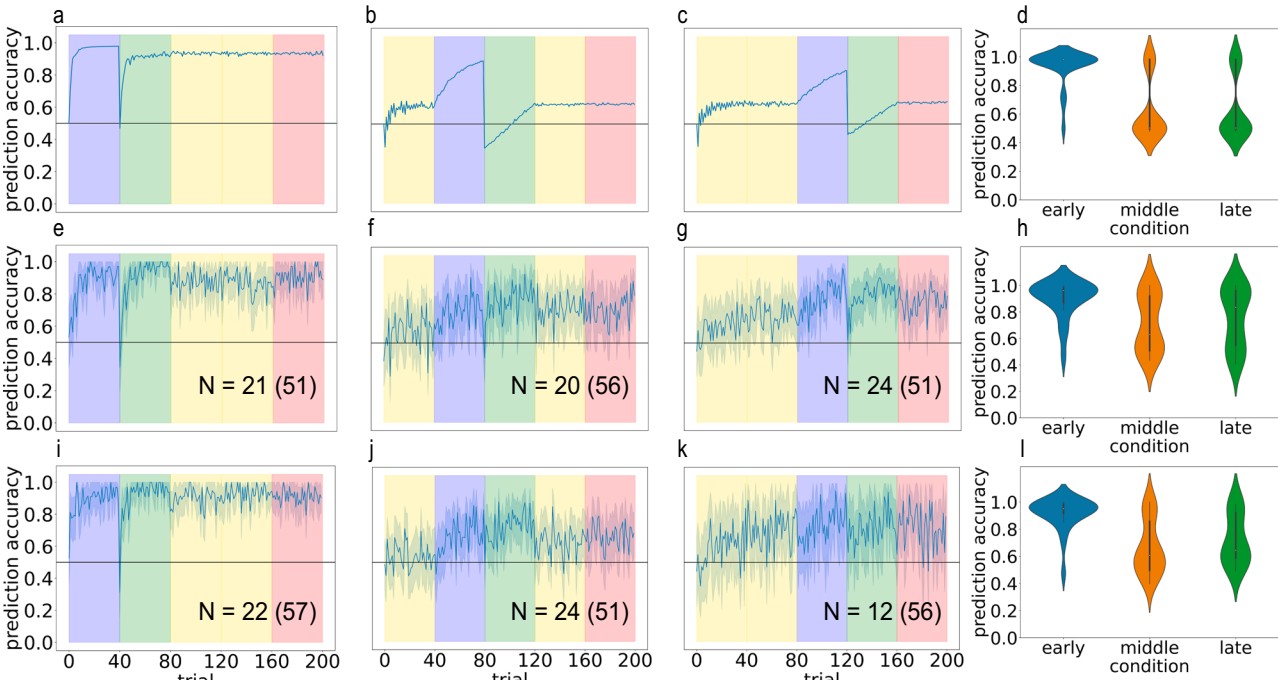

**Fig. 10 | Inserted blocks experiments confirm the prediction that inserting blocked training early during learning results in better performance compared to inserting blocked training at the end. a–c** Model predictions on early, middle and late curricula, respectively. **d** Violin plots of test accuracy predicted by the model on each curriculum. **e–g** Human accuracy over time on early, middle and late curricula, respectively. N indicates the number of participants included in the final analysis; the number in parentheses indicates the total number of participants before exclusion (see attention check and exclusion criteria in Methods). **h** Violin plots of human test accuracy confirm the prediction that early curriculum leads to better performance than middle and late. **i–l** Replication of (**e–h**) on independent sample of human participants. Error ribbons indicate ± 1 standard error.

statistically significant difference in performance (t(42) = 0.634, p = 0.529, d = 0.192, 95% CI = [−0.0926, 0.177]). This was also replicated in an independent sample (t(34) = 0.827, p = 0.414, d = 0.292, 95% CI = [−0.0826, 0.196]). These results were qualitatively similar without any participant exclusion. The early condition was better than both the middle condition (original experiment: t(105) = 2.70, p = 0.00814, d = 0.522, 95% CI = [0.0288, 0.189]; replication: t(106) = 1.89, p = 0.0612, d = 0.365, 95% CI = [−0.00366, 0.157]) and late condition (original experiment: t(100) = 2.07, p = 0.0407, d = 0.411, 95% CI = [0.00386, 0.174]; replication: t(111) = 3.53, p < 0.001, d = 0.665, 95% CI = [0.0569, 0.202]), and the middle and late conditions did not show a statistically significant difference in performance (original experiment: t(105) = 0.508, p = 0.612, d = 0.0984, 95% CI = [−0.0569, 0.0961]; replication: t(105) = −1.66, p = 0.100, d = −0.321, 95% CI = [−0.117, 0.0104]). This supports the interpretation that interleaved training induces inaccurate segmentation of the stories that makes subsequent learning difficult.

## Discussion

How are schemas learned and represented? In this paper, we used a multi-schema learning task and found that learning was best under a blocked compared to an interleaved training curriculum. The poor performance of our participants in the interleaved condition is especially remarkable given that the interleaved task is easily learnable by an off-the-shelf recurrent neural network. Our modeling studies suggest that these curriculum effects can be explained by Bayesian latent cause models. These models accurately carve the world at its joints after blocked training (where shifts in the generative model align with spikes in prediction error) but not after interleaved training (where prediction errors do not align as clearly with shifts in the generative model).

Importantly, this framework does not predict that interleaved learning will always be worse than blocked learning: In situations where prediction errors are below the "splitting threshold" (in both the interleaved and blocked conditions) then interleaved learning can be better than blocked, due to catastrophic interference in the blocked condition (as we show in our LSTM simulation). This provides a potential way of reconciling our results with a body of literature showing better learning given interleaved (vs. blocked) training[9,10,24]. This does, however, raise the question of what features of our paradigm put it into the "large prediction error" regime associated with splitting (and good performance in the blocked learning condition). While we can only speculate at this point, it may be that our use of a "stop and ask" paradigm where participants have to explicitly predict the next state (before seeing that state) is important for generating prediction errors large enough to cause splitting.

### Curriculum effects on attention

We do not wish to claim that splitting is the only factor that can lead to superior learning in blocked vs. interleaved curricula. The recent literature exploring curriculum effects on inductive category learning provides a useful illustration of this point. While many category learning studies have found that interleaved training results in better learning compared to blocked training (e.g. ref. 9–11), other studies have found improved learning in blocked compared to interleaved curricula (e.g., ref. 25–29). Some of these conflicting results have been explained in terms of the Attention Bias Framework (for a review see[30]), according to which different curricula alter what features are attended to and therefore encoded: While interleaved curricula help category learning when exemplars of different categories are similar to one another (by making it easier to notice subtle differences between the categories), blocked curricula are better when exemplars from a given category are dissimilar from each other (by making it easier to notice subtle commonalities between exemplars within a category; see also[31]). Another recent set of studies[32] found an advantage for blocked over interleaved learning in a category-learning task; furthermore, they provide evidence that blocked training results in a more "factorized" representation of stimulus dimensions. A follow-up study[33] showed that a combination of "sluggish" units whose activity reflects the recent history of stimuli, together

with a Hebbian learning rule, is sufficient to allow neural network models to develop these "factorized" task representations (see also[34]). Taken together, these studies provide clear evidence that the training curriculum can affect how participants (and models) attend to / represent high-dimensional visual stimuli, which in turn can affect learning.

We think that it is unlikely that attentional factors played a role in giving rise to the curriculum effects mentioned here: While our stimuli were complex in the sense that they unfolded over time according to the transition probabilities in the Markov chain, the features defining the individual states in the chain were very obvious (e.g., it was clear when participants were "waiting patiently" vs. "cutting the line" in the text narratives). Furthermore, results from our "explicit interleaved" condition (Fig. 4) show that interleaved learning continued to be much worse than blocked learning even in the situation where participants' attention was repeatedly drawn to the schema-defining location feature. As such, we hypothesize that the studies reported here involve a complementary (and non-mutually-exclusive) mechanism from the category-learning studies mentioned above. A potential future direction of research would be to understand how attentional and representational-splitting accounts interact. For example, manipulations that help orient participants to the features that define a category or schema may increase the odds that splitting will occur when the category or schema changes.

### Limitations

We found a benefit of blocked over interleaved schema learning across multiple direct and conceptual replications, but more work is needed to identify the necessary and sufficient features for obtaining the pattern of results reported here. Earlier, we mentioned the potential contribution of forcing participants to explicitly predict the next state – if we took this explicit prediction demand away, would the prediction errors still be large enough to cause splitting? Another feature of our current paradigm is that we do not disclose the true number of latent causes (schemas). If we explicitly informed participants that there were two schemas, would that improve schema inference in the interleaved condition? This gets at the question of how "cognitively penetrable" the splitting process is – i.e., is it under some degree of explicit control and thus modifiable by information about the true number of latent causes, or does it proceed automatically?

In addition to considering modifications to the experiment instructions, one can also explore variants of the schemas themselves. The environments that we used were designed to have maximally confusable states between schemas: The states produced by the two Markov chains were associated with identical observations (except in the "explicit interleaved" condition), and the transition structures of these two schemas were mirror-opposite to equate the probability of being in any given state under both schemas. Future work could investigate other designs. For example, if states were not aliased, there should be no need for prediction-error-based splitting, as the differences in the observations linked to each state should sufficiently separate representations of these states.

Another future direction is to improve our explanation of individual differences: Why is it that some participants seem to learn in the interleaved condition but others do not? As mentioned earlier, one promising approach would be to estimate individual differences in key parameters governing latent cause inference (e.g., the concentration parameter, $\alpha$), using a separate set of tasks; we could then see whether these estimated parameters relate to interleaved learning performance in a way that aligns with the predictions of our Bayesian model (for other work exploring individual differences in latent cause parameters, see, e.g., ref. 35–37).

There is also work to be done in reconciling the Bayesian model with the LSTM neural network model used here. The reason that our best-fitting Bayesian model ("Simulation 3") predicts failure-to-learn in the interleaved condition is because stimuli generated by chain A and chain B get lumped together into a single latent cause. While it is clear why this "lumping" causes impaired performance in our Bayesian model (which has a very simple representation for each latent cause – basically, just a table of observed transitions), it is less clear why this lumping would impair performance in a

model that represents latent causes in a more sophisticated way. Indeed, our LSTM simulations showed excellent learning in the interleaved condition despite their use of a unified ("unsplit") network to represent the stimuli – this is because of the well-known ability of recurrent neural networks to learn context-sensitive temporal structure[22]. Put another way: If we want to explain interleaved learning failure in terms of failure-to-split, we need to explain how an unsplit neural network could fail to learn in the interleaved condition. One promising possibility is that our brains incorporate priors on learning (not present in our simple LSTM) that generally help learning, but can also harm learning when environmental statistics do not accord with the priors. As mentioned earlier, a prior study[33] relied on "sluggish" units to explain blocked superiority effects in category learning (basically, the sluggishness results in smearing of activation across distinct categorization rules, impeding performance when there are frequent switches), and this sluggishness property also has strong similarities to the sticky CRP prior used by our Bayesian model. Future work can assess whether adding some kind of sluggishness prior to a recurrent neural network would help to explain interleaved learning failure in our task.

Our Bayesian model might also benefit from the addition of learning principles that are not (yet) present in the model. For example, in our Bayesian model, learned schemas are given completely distinct representations; this means that there can be no generalization between schemas. However, we note that – in the blocked condition – human data shows some benefit of learning the first schema on learning of the second schema. This should not be so surprising, given that there are some commonalities between both schemas: For example, once the first schema has been learned, learning the second schema simply amounts to realizing that the transitions are inverted. Adding compositionality to the model could help to explain these results[38]. Also, the Bayesian model currently assumes that participants can ignore unpredictable transitions when making inferences about latent causes, but it does not actually explain this. Future work could add a learning mechanism to the model that estimates predictability and factors these predictability estimates into inferences, as hypothesized by normative models of learning[39,40]. The basic idea is that learning should only occur for predictable (i.e., learnable) transitions. Because predictability is a latent variable, a more sophisticated model would attenuate updating in proportion to inferred unpredictability.

Finally, another future direction is to use fMRI and other neural measures to seek converging evidence for the claims made here. While our model predicts that participants will (mostly) learn a two-schema solution in the blocked condition, other solutions are possible; in particular, participants could learn a four-schema solution where each of the four "paths" in Fig. 1 is assigned its own schema. Behavioral prediction-accuracy data can not, on its own, distinguish between the two-schema and four-schema possibilities (since they both support accurate prediction). However, fMRI data could potentially be used to tease them apart – e.g., one could use representational similarity analysis[41] to determine whether there is a pattern that is shared by all states that are part of chain A and not by states part of chain B (indicative of a two-schema solution), and/or whether there is a pattern that is shared by all states on the left "path" of chain A but not the right "path" of chain A (indicative of a four-schema solution).

## Concluding remarks

In summary, the results presented here provide strong support for splitting models[13,15,16,19]. When an environment is generated by different latent causes, splitting knowledge about these latent causes into distinct representations can provide critical benefits for an adaptive system that builds a collection of schemas across its lifespan, improving the stability of the system by allowing it to hold onto existing knowledge even when training occurs in blocks.

## Methods

### Text narrative experiments: ethics approval
All participants provided written informed consent prior to the experiment. The experiment protocol and the consent forms were approved by the

Institutional Review Board of Princeton University (protocol number 10374).

### Text narrative experiments: participants
Participants were recruited on Amazon Mechanical Turk (AMT). We aimed to collect about $N = 50$ per experiment. However, because of fluctuations on AMT servers, multiple participants sometimes initiated the experiment at the same time, so the actual numbers varied. Furthermore, although we recruited approximately 50 participants per condition, many were excluded for failing the attention check (see below). The final numbers of participants for each condition were: blocked $N = 33$ (50), interleaved $N = 39$ (55), blocked replication $N = 26$ (63), interleaved replication $N = 24$ (56), explicit_interleaved $N = 31$ (52), early inserted blocks $N = 21$ (51), middle inserted blocks $N = 20$ (56), late inserted blocks $N = 24$ (51), early inserted blocks replication $N = 22$ (57), middle inserted blocks replication $N = 24$ (51), late inserted blocks replication $N = 12$ (56); the number inside parentheses represents the number of recruited participants before the exclusion criterion was applied (see below). The numbers of male and female participants were not recorded for these experiments.

The replications of the blocked and interleaved studies were preregistered (date of preregistration: April 7, 2019); the preregistration can be viewed here: https://osf.io/ebxfa/.

### Text narrative experiments: attention check and exclusion criterion
To ensure a minimal amount of attention was paid to the experiment, we included attention checks in our experiment. These attention checks were in the form of questions, of the same form as those that were asked during the ongoing task, that participants should be able to answer correctly if they are reading the text on the screen. The following would appear on the screen: "Alice walked into the coffeeshop", "What happens next", and the two response choices would be "Alice ordered cake" (correct), "Bob ordered cake" (incorrect). Approximately 25% of questions were of this form (see *Data analysis and statistics* section below). Participants who did not get 90% of these questions correct were excluded from our main analyses; however, all results were qualitatively the same when the full set of participants was included – these "unfiltered" results are reported in the paper for key analyses (see, e.g., Supplementary Fig. 3).

### Text narrative experiments: payment structure and incentive bonus
Every participant was paid a minimum of $6 and a maximum of $8 for completing the experiment. To incentivize participants to pay attention, we instructed them that they would be paid $4 for completion, with the possibility of earning another $4 depending on how many correct responses they made. In truth we always paid a minimum of $2 in bonus, and we gave an extra $2 of bonus depending on the number of attention check questions they got correct.

### Text narrative experiments: procedure
Participants read 200 stories at their own pace, one sentence at a time. After reading about a given event in the story (e.g. "Alice waited in line patiently") participants pressed the spacebar on their keyboard to get the next sentence. On some deterministic transitions (explained below), after pressing the spacebar, participants were given a two-alternative forced choice (2AFC) prediction probe. This probe was of the following form: "Alice waited in line patiently; What do you think will happen next?", they could then indicate with their left or right arrow key one of two responses "While ordering tea, Alice noticed the barista's new mustache", or "Before ordering cake, Alice took a quarter from the tip jar". After using the right or left arrow key to respond, the next event would show up on the screen: "Before ordering cake, Alice took a quarter from the tip jar". There was a 2500ms enforced minimum time for each sentence to discourage people from rushing through the experiment. At the end of each story, a screen with the following words appeared: "NEW STORY". Each experiment lasted about 45 minutes.

## Text narrative experiments: stimuli

The stories read by participants were generated from chains, where each chain state corresponds to an event in the story. Each event was communicated with a single sentence. Below are the sentences used in all stories:

- State 0: "[subject.name] wants to go out today."
- State 1: "[subject.name] decides to go to the Jungle Brew House."
- State 2: "[subject.name] decides to go to the Deep Ocean Cafe."
- State 3: "Today, [subject.name] waited in line patiently."
- State 4: "[subject.name] was impatient, and decided to cut the line."
- State 5: "While ordering tea, [subject.name] noticed the barista's new mustache."
- State 6: "Before ordering cake, [subject.name] took a quarter from the tip jar."
- State 7: "[subject.name] then sat by the window, and read a book for hours."
- State 8: "After ordering, [subject.name] stole a salt shaker and left."
- State 9: "That is all that is remembered."
- For each story, a different subject name was filled in to the slot indicated by [subject.name]. This was done to ensure that each story was slightly different from every other story, and to facilitate the attention checks described above.

## Text narrative experiments: story chains

Each story was generated by one of two chains. Both chains began by generating sentence 0. From there, chain A transitioned to sentence 1, while chain B transitioned to sentence 2. From there, both chains transitioned to sentences 3 or 4 with 50% probability. The critical difference was in the next two transitions: While chain A took the transitions 3-5-7-9 or 4-6-8-9, chain B took the mirror opposite transitions 3-6-7-9 or 4-5-8-9. In sum:

Chain A generated: 0-1-3-5-7-9 or 0-1-4-6-8-9

Chain B generated: 0-2-3-6-7-9 or 0-2-4-5-8-9

This transition structure was used to ensure the probability of states 1-8 unconditioned by chain was 50%; however, if conditioned on chain, the next state could be deterministically predicted (except for the unpredictable transition into the 3/4 states).

## Text narrative experiments: curriculum manipulation

For convenience, we refer to the first 160 stories as the training phase and the remaining 40 stories as the test phase; however, this division was not made explicit to participants.

During the training phase, the chain from which the story was generated was specified by the curriculum condition. In the blocked condition, stories were generated in blocks of 40 from the same chain, with adjacent blocks being generated by different chains (40 from chain A, then 40 from chain B, then 40 from chain A, then 40 from chain B). In the interleaved condition, each adjacent story was generated by a different chain, so that chains perfectly alternated for adjacent stories. For the inserted blocks experiments, there were three training curricula – early, middle and late – with the following structure.

- Early (A B I I): 40 stories from chain A, 40 stories from chain B, 80 stories interleaved
- Middle (I A B I): 40 stories interleaved, 40 stories from chain A, 40 stories from chain B, 40 stories interleaved;
- Late (I I A B): 80 stories interleaved, 40 stories from chain A, 40 stories from chain B.

To make for a fair comparison between different training conditions, the final 40 (test phase) stories were generated by the same random curriculum in all conditions; that is, for each of the final 40 stories, chain A or B was chosen with 50% probability each.

## Text narrative experiments: statistics and reproducibility

Each participant responded to 2 two-alternative forced choice (2AFC) questions per story. These questions probed the following (deterministic)

transitions: the transition from states 3/4 to states 5/6; and the transition from states 5/6 to states 7/8. For each of these 2AFC questions, there was a 25% probability that the standard next-state-prediction question would be replaced with an attention-check question that tested the participant's memory for the name of the main character of the story.

While participants were exposed to a total of 200 stories, the first 160 stories were designated as the training period. Analyses focused on the remaining 40 stories, designated as the test period. Note that participants experienced no discontinuities between story blocks or the train-test transition.

The test accuracy for each participant was calculated by taking a mean of the correct 2AFC responses across the 2 questions for each of the 40 test stories. Since all transitions that were probed in this fashion were deterministic, a correct response corresponds to the true transition on the generative process (see Fig. 1). For all experiments, we used two-tailed t-tests to evaluate for statistical significance, and we used Cohen's d to index effect size.

## Animated narrative experiments: ethics approval

All participants provided written informed consent prior to the experiment. The experiment protocol and the consent forms were approved by the Institutional Review Board of Princeton University (protocol number 7883).

## Animated narrative experiments: participants

This experiment collected three independent samples of participants. All participants gave informed consent and were financially compensated for their participation. AMT participants were given $7 base pay per hour, plus a $3 "performance bonus" that they received if they completed both sessions of the experiment. In-person participants were paid at a rate of $12 per hour.

Group 1 (AMT). 82 participants recruited from AMT participated in this study. Of these 82 participants, 43 were randomly assigned to the blocked group (27 included for final analysis; 13 men, 13 women, and one participant who preferred not to answer, based on self-report; mean age 35.4), and 39 to the interleaved group (21 included for final analysis; 13 men and 8 women, based on self-report; mean age 32.4). In total, 34 participants were excluded: 27 did not finish the first day of the study, 3 did not finish the second day of the study, 3 reported to not hear the audio well enough, and 1 failed the general attention check (see below).

Group 2 (AMT). 191 participants recruited from AMT participated in this study. Of these 191 participants (127 included in analysis), 96 were randomly assigned to the blocked group (66 included in analysis; 33 men and 33 women, based on self-report; mean age 36.4), and 95 to the interleaved group (61 included in analysis; 32 men and 29 women, based on self-report; mean age 37.2). In total, 64 participants were excluded because of not finishing the entire study.

Group 3 (onsite). 30 participants recruited from Princeton University participated in this study. 15 participants were randomly assigned to the blocked group (7 men and 8 women, based on self-report; mean age 20), 15 to the interleaved group (5 men and 10 women, based on self-report; mean age 20). The experiment was conducted at a behavioral lab on the Princeton University campus.

## Animated narrative experiments: procedure

This experiment was conducted over the course of two days. On Day 1, participants were exposed to 24 fictional animated videos of wedding ceremonies. Ceremonies were presented in 4 blocks of 6 ceremonies each, with 2 min breaks between blocks. The order of the ceremonies (i.e. training curriculum) was manipulated across two conditions: Participants in the blocked condition received 12 stories generated from the A chain, followed by 12 stories generated from the B chain (i.e. A x 12 – B x 12). Participants in the interleaved condition received a story from the A chain, followed by a story from the B chain, and then back to the A chain, and so on (i.e. A-B-A-B…).

After the second ritual (state 3 or 4) and third ritual (state 5 or 6) of the first and last wedding of each block, participants were probed with 2AFC prediction questions (described below). That is, they were probed on the transitions leading out of states 3–6, inclusive. They had a 4 sec to answer; if the time expired, the response was marked as incorrect. For example, after watching a couple celebrate around the campfire, participants would get a screen that asked "What do you think will happen next?", with two options given on the left and right side of the screen ("drop coin in bowl" and "hold torch") corresponding to the two possible next rituals.

To keep participants engaged, they were also probed with 6 2AFC episodic detail questions (described below) after each ceremony (presentation time is 2 sec per question, no ITI). For example, after the termination of the ceremony, two objects would appear on the screen along with the question "what object appeared during this ceremony?". Episodic detail questions were used as attention checks for participants in the AMT samples (see Attention Check and Exclusion Criteria below), and not further analyzed in the current study.

On Day 2, participants in both the blocked and the interleaved condition were exposed to a novel set of 6 North and 6 South ceremonies in the following order N-N-S-S-N-N-S-S-N-N-S-S; this alternating structure was put in place for reasons unrelated to the work presented here (the audiovisual experiments included additional episodic memory tests at the very end of the experiment session – for these episodic memory analyses, we wanted to make sure that we had a roughly equal number of temporally-adjacent "same-schema" and "different-schema" weddings; data from these tests will be described in another paper). While watching, participants again received 2AFC prediction questions during the first and last ceremony they saw. As in Day 1, they received these prediction questions right before the start of the second and third rituals, and they also received the 6 episodic detail questions at the end of each episode.

At the end of Day 2, participants received 8 schema questions. These were similar to the prediction questions in that they probed for knowledge of the transition structure. However, instead of being presented in the middle of the wedding videos, they occurred at the end of Day 2. Schema questions were FC questions that explicitly asked participants about the transition structure, e.g., "if a couple from the North just celebrated around a campfire, what will most likely happen next?" (see full list of schema questions below). All analyses reported in the Results section were done on these schema questions.

## Animated narrative experiments: attention check and exclusion criteria

A predefined exclusion threshold was set whereby participants who got less than 50% correct in the episodic detail questions correct were excluded from final analyses. Only one participant was excluded for this reason (as noted above, the most frequent reason for excluding participants was that they did not complete the experiment).

## Animated narrative experiments: stimuli

Audiovisual narratives were generated using Unity software (https://unity3d.com). Using Unity, we created clips depicting rituals in a marriage ceremony. The audio was imported into Audacity for further processing before it was integrated with the video. The audio files were imported together, and then normalized. Remaining audio processing was done in iMovie, using the auto sound editor, background noise reducer, and the equalizer ("voice enhance"). Furthermore, the volume of each video was adjusted to a subjectively good level.

Each ritual clip corresponded to a state on a Markov chain. Therefore, a draw from this chain defines a sequence of rituals that makes up a marriage ceremony. Each ceremony consisted of a series of rituals that lasted a total duration of 2 minutes each. Below is a list of rituals used to put together a marriage ceremony, along with the corresponding state numbers from Fig. 1:

- start of the wedding (State 0)
- celebrate around campfire (State 3)
- plant a flower (State 4)
- drop coin in bowl (State 5)
- hold a torch (State 6)
- break an egg (State 7)
- draw a painting (State 8)
- receive gifts (State 9)

Each wedding had different protagonists (names and faces). Each wedding also had a different set of objects present in the scene (e.g., specific gifts received, specific paintings, specific pictures placed on the egg, etc.). These served as episodic details that were probed in the episodic detail questions below (see procedure).

The wedding videos can be viewed at: https://osf.io/u3cfr/.

## Animated narrative experiments: chain structure

The underlying chain structure from which these wedding ceremony paths were generated was identical to the chain structure in every other experiment and simulation reported here. As before there were two chains with aliased states and mirror opposite transition structures. While in the first experiment Jungle Brew House versus Deep Ocean Cafe indicated whether chain A or chain B generated the story, here it was whether the marriage was between couples from the North versus South side of a fictional island. When a wedding couple came from the North, the rituals performed during their wedding followed one of the following two paths:

North path-1: start of the wedding (state 0) - celebrate around campfire (state 3) - drop coin in bowl (state 5) - break an egg (state 7) - receive gifts (state 9)

North path-2: start of the wedding (state 0) - plant a flower (state 4) - hold a torch (state 6) - draw a painting (state 8) - receive gifts (state 9)

When a couple came from the South, the rituals performed during their wedding followed one of the following two paths:

South path-1: start of the wedding (state 0) - celebrate around campfire (state 3) - hold a torch (state 6) - break an egg (state 7) - receive gifts (state 9)

South path-2: start of the wedding (state 0) - plant a flower (state 4) - drop coin in bowl (state 5) - draw a painting (state 8) - receive gifts (state 9)

At the start of each ceremony, participants saw a text-cue that indicated whether the following ceremony is from a North-couple or South-couple.

## Animated narrative experiments: schema questions

All analyses done on the animated narrative experiments were done on the following schema questions that happened at the end of Day 2.

The schema questions tested participants' knowledge of the ritual transition structure of ceremonies from the North versus South. The questions were displayed in the following format: e.g. "If a couple from the North just planted a flower, what is the most likely ritual to happen next?". In the first animated narrative experiment, participants were provided with a 2AFC of the two rituals that might follow. There were 8 such questions:

- North couple, right after celebrating around a campfire
- North couple, right after planting a flower
- North couple, right after dropping a coin in a bowl
- North couple, right after holding a torch
- South couple, right after celebrating around a campfire
- South couple, right after planting a flower
- South couple, right after dropping a coin in a bowl
- South couple, right after holding a torch

In the second and third animated narrative experiments, participants were given a similar set of questions at the end of Day 2. However, instead of 2AFC, participants were asked to allocate 100% of their confidence across all six of the wedding rituals (campfire, flower, coin, torch, egg, painting). It was explained to them that they were free to either put the entire 100% on one answer or split it across multiple answers.

## Animated narrative experiments: statistics and reproducibility

Analyses were done on the schema questions administered at the end of Day 2. These questions probed for knowledge of the transition structure: They were FC questions that explicitly asked participants about the transition structure, e.g., "if a couple from the North just celebrated around a campfire, what will most likely happen next?" (see Schema Questions, above). As in the text narratives experiment, participants in Experiment 1 provided a response in the form of a 2AFC, where the options corresponded to the true next state and the next state corresponding to the alternative chain. Participants in Experiments 2 and 3 were given a response set containing 6 alternatives, corresponding to every possible state in the chain. They were instructed to allocate 100% across all 6 alternatives. For all experiments, we used two-tailed t-tests to evaluate for statistical significance, and we used Cohen's d to index effect size.

Note that, since the data showed a deviation from normality as indicated by significant Shapiro-Wilk tests ($P < 0.05$), we also ran Mann-Whitney $U$ tests to test the difference between the blocked and the interleaved group (with $p < 0.05$ used as a criteria for significance using two-tailed tests). All results were qualitatively the same. We also used Levene's test to assess the equality of variances and found no statistically significant differences in variances across conditions (i.e. $p > 0.05$).

## Long-short term memory simulations: task generation

The goal was to keep these simulations as close to the behavioral experiments as we possibly could. Accordingly, the stimuli were generated according to the same chains used in the behavioral experiments. There were two chains with mirror opposite transition structures, and with aliased states: That is, the states of the two chains produced the same observation, while the transitions from both chains were different. However, instead of representing events in a story, each state of this chain is encoded as a one-hot vector representation of that chain state. Therefore, a pass through this chain generated a sequence of one-hot vectors that can be interpreted as a story. This sequence of one-hot vector representations gets sequentially fed through the model.

## Long-short term memory simulations: forward propagation

The network receives a sequence of one-hot inputs, each corresponding to a given state of the event chain. These one-hot vectors are projected through a fully connected linear layer, resulting in the distributed representation embedding for that state. This embedding is then passed as input to a LSTM cell: After updating its internal state, the LSTM cell produces an output vector. The output from the LSTM cell is passed through a final fully connected layer, where each output unit of this layer corresponds to a given state on the chain. These output activations are passed through a softmax activation function, which can be interpreted as the network's prediction for the next state in the form of a probability distribution over all possible next states.

## Long-short term memory simulations: training and evaluation

The network was trained using standard backpropagation with an Adam optimizer[42]. Training data were generated by first generating a sequence draw from the chain. Every element of that sequence, except for the last, gave rise to a training sample. The training samples consisted of a current state, provided as input, and a true next state, provided as a label. We were interested in testing for catastrophic interference (CI) between chains. To do this, we evaluated the network after each training epoch by freezing the weights and providing it with input sequences corresponding to both event chains. This allowed us to investigate what happens to the knowledge representation of one chain while being trained to learn the other.

## Bayesian model: generative process

Our Bayesian simulations involved defining a generative process that matches the structure of the experiments, and inverting that process to derive the optimal inference strategy. Effectively, the experiment consists of two latent states (corresponding to the different Markov chains) that condition the observed state transitions. Thus, at each time step $t$, a new state $s_t \in \{1, \ldots, S\}$ is sampled from a transition distribution $T(s_t|s_{t-1}, z_{t-1})$ conditional on the current state $s$ and current chain $z_t$. We assume that the participant has no expectation about the number of possible chains that are being used to generate the environment. Therefore, we used a sticky Chinese Restaurant Process[18,23] as the prior for the chain sampling distribution:

$$P(z_t = k|\mathbf{z}_{1:t-1}) \propto \begin{cases} N_{tk} + \beta\delta[z_{t-1}, k] & \text{if } k \text{ is old} \\ \alpha & \text{if } k \text{ is new} \end{cases} \quad (1)$$

where $N_{tk}$ is the number of times schema $k$ has been sampled prior to $t$, $\alpha \geq 0$ is a concentration parameter, and $\beta \geq 0$ is a stickiness parameter weighting a delta function on the previous schema. Note that "new" here refers to the *next* unused schema (e.g., if five schemas have already been used, then the prior probability of the sixth schema is given by the "new" case in the above formula, and the prior probabilities of schemas seven and above are set to zero).

We also assume that the participant does not know the distribution from which states are drawn. Thus the transition distribution is sampled from a symmetric Dirichlet distribution:

$$T(\cdot|s, k) \sim \text{Dir}(\lambda). \quad (2)$$

The sparsity parameter $\lambda \geq 0$ controls the shape of the prior. When $\lambda = 1$, the prior is uniform. When $\lambda < 1$, the prior has symmetric peaks at 0 and 1, meaning that the deterministic transition distributions are favored. When $\lambda > 1$, the prior is peaked at $1/S$, favoring a uniform transition distribution.

## Bayesian model: inference

Having defined the generative process for the environment, we simply invert the model to perform schema inference according to:

$$P(z_t|\mathbf{s}_{1:t}, \hat{\mathbf{z}}_{1:t-1}) \propto P(s_t|\mathbf{s}_{1:t-1}, z_t, \hat{\mathbf{z}}_{1:t-1})P(z_t|\hat{\mathbf{z}}_{1:t-1}), \quad (3)$$

where the second term on the right hand side is the sticky CRP given above, and the first term is the marginal likelihood, obtained by marginalizing over the transition distribution:

$$P(s_t = j|s_{t-1} = i, \mathbf{s}_{1:t-2}, z_t = k, \hat{\mathbf{z}}_{1:t-1}) = \frac{\lambda + M_{tkij}}{S\lambda + \sum_{j'} M_{tkij'}}, \quad (4)$$

where $M_{tkij}$ is the number of $i \to j$ transitions observed prior to $t$ when the active schema was $k$; put another way, the model uses $M_k$ to record, in a lossless fashion, the history of transitions that were observed when schema $k$ was active. Exact Bayesian inference over the transition distribution $T$ and the schema history $\mathbf{z}$ is intractable, because the number of possible schema histories explodes exponentially. To address this, we adopt the "local maximum *a posteriori*" (local MAP) approximation. The point estimate for the schema history is updated as follows:

$$\hat{z}_t = \underset{k}{\text{argmax}} \, P(z_t = k|\mathbf{s}_{1:t}, \hat{\mathbf{z}}_{1:t-1}). \quad (5)$$

In other words, we "freeze" the schema history to be the locally optimal point estimate (i.e. also referred to as the "active schema").

The Bayesian optimal predictive distribution would be obtained by marginalizing across schemas:

$$P(s_t|\mathbf{s}_{1:t-1}) \approx \sum_k P(s_t|\mathbf{s}_{1:t-1}, z_t = k, \hat{\mathbf{z}}_{1:t-1})P(z_t = k|\hat{\mathbf{z}}_{1:t-1}). \quad (6)$$

Marginalizing across schemas can be computationally (and thus psychologically) cumbersome, especially as the number of inferred schemas

increases. This led us to default to the less-cumbersome (and thus more-psychologically-plausible) approximation wherein only the currently active schema is used to make predictions[43]:

$$P(s_t|\mathbf{s}_{1:t-1}) \approx P(s_t|\mathbf{s}_{1:t-1}, z_t = \hat{z}_{t-1}, \hat{\mathbf{z}}_{1:t-1}). \quad (7)$$

Note that, for completeness, we also ran a variant of Simulation 3 using the Bayesian optimal approach. The results of this simulation are shown in Supplementary Fig. 9; overall, the fits provided by the Bayesian-optimal approach were qualitatively similar but quantitatively slightly less good.

### Bayesian model: simulation

Putting the above equations together, the control flow of our simulations are given by the following algorithm: On each trial, the model observes the current state of the environment. The model uses the currently active schema to make a prediction about the next state of the environment. The model is then given the correct next state so it can update the likelihood value for all of the schemas in its library. Next, the model computes a posterior probability for all of the schemas in its library (including the probability that a new, not-previously-used schema should be used). Lastly, the model takes the argmax of the posteriors. If the current schema has the maximal posterior probability, that schema stays active; if another schema (including the new, not-previously-used schema) has the maximal posterior posterior probability, the model switches to that schema.

### Bayesian model: accuracy of model prediction

Each schema of the model contains a transition matrix M(i,j) that encodes the number of $i \rightarrow j$ transitions. On each trial the environment emits an observation corresponding to one of 9 states from Fig. 1. Note that the two schema-identifying states (states 1 and 2) are treated the same as the other 7 states. As a minor note, the state labeled as state 9 in Fig. 1 was never shown to the model (this state occurs at the end of every story and participants were never asked to predict it, so it was simpler to leave it out).

The model prediction is given by the row M(i) - i.e a 9-dimensional vector, each entry of which corresponds to one of the states in the environment. This vector is the likelihood of each outgoing transitions from state $i$. To issue a prediction, this likelihood vector is passed through a softmax (temperature = 4), which gives a probability distribution over transitions. Note that this vector encodes the probability distribution over 9 possible states, but human participants were given a 2AFC and would thus only have to evaluate the posterior odds of those two states. To match the model to humans, we took the two transitions allowed from the current state and normalized their probabilities. The model accuracy for each trial is then equal to the probability the model assigns to the transition that actually occurred on that trial.

### Bayesian model: parameter gridsearch

To find the best-fitting parameters, we randomly sampled different parametrizations of the model within a uniform cube in parameter space, and evaluated the model accuracy against human accuracy using mean squared error (see below). For each parameter set $(\alpha, \lambda, \beta)$ drawn from $\alpha \sim U[0.001, 100]$, $\beta \sim U[0.001, 100]$, $\lambda \sim U[0.001, 1.2]$, where $U[A, B]$ denotes the uniform distribution between $A$ and $B$, $N = 100$ random seeds of the model were simulated on each of the conditions (Blocked, Interleaved). For each such model instance, we computed an accuracy over trials, and averaged across model seeds to get an accuracy for each condition.

These model accuracy traces were then compared to human accuracy by taking the mean squared error (MSE) deviation between model accuracy and human accuracy. The MSE was computed as $\sum_t (model\_acc_t - human\_acc_t)^2$, where $t$ indexes story number (1 through 200). We computed this MSE score separately for the blocked and interleaved conditions (using human data from the original blocked and interleaved experiments) and averaged these values together to get the final MSE that was used to evaluate a particular parameter configuration.

### Bayesian model: variants

We explored three model variants, each of which was given its own grid-search. Simulation 1 in the main text used the "base" variant of the model. Simulation 2 was the same as Simulation 1 except we had the model skip over the unpredictable transition (from states 1 and 2 to states 3 and 4) when inferring which latent cause should be active. Simulation 3 was the same as Simulation 2 except we added variability across seeds (within a parameter configuration). In Simulation 3, for each model seed, we sampled a different value of the concentration parameter from a normal distribution. The standard error of this distribution was fixed to 0.3 and the mean was discovered by the gridsearch process described above.

The best-fitting parameter values for the three simulations were:
- Simulation 1 (MSE = 0.0919): $\alpha = 4.775$, $\beta = 96.792$, $\lambda = 0.051$
- Simulation 2 (MSE = 0.0406): $\alpha = 1.703$, $\beta = 1.848$, $\lambda = 0.211$
- Simulation 3 (MSE = 0.0371): $\alpha = 3.604$, $\beta = 5.057$, $\lambda = 0.436$.

### Reporting summary

Further information on research design is available in the Nature Portfolio Reporting Summary linked to this article.

### Data availability

Data are available at: https://github.com/PrincetonCompMemLab/blocked_training_facilitates_learning.

### Code availability

Analysis code and simulation code are available at: https://github.com/PrincetonCompMemLab/blocked_training_facilitates_learning. The specific version of the code that was used for the analyses reported here is available at: https://doi.org/10.5281/zenodo.10695055.

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

## Acknowledgements

This work was funded by ONR MURI grant N00014-17-1-2961 awarded to KAN and SG and NWO Rubicon grant (446-17-009) awarded to SHPC. The funders had no role in study design, data collection and analysis, decision to publish, or preparation of the manuscript.

## Author contributions

Andre O. Beukers: Methodology, Software, Validation, Formal analysis, Investigation, Resources, Data Curation, Writing - Original Draft, Visualization, Conceptualization. Silvy H.P. Collin: Methodology, Software, Validation, Formal analysis, Investigation, Resources, Data Curation, Writing - Review and Editing, Conceptualization. Ross Kempner: Software, Validation, Formal Analysis, Data Curation, Visualization. Nicholas T. Franklin: Conceptualization, Validation, Investigation. Samuel J. Gershman: Conceptualization, Supervision, Funding acquisition, Project administration, Writing - Review and Editing. Kenneth A. Norman: Conceptualization, Supervision, Funding acquisition, Project administration, Writing - Review and Editing.

## Competing interests

The authors declare no competing interests.
