## [Peer Review File · Communications Psychology]

15th Sep 23

Dear Professor Norman,

Thank you for your patience during the peer-review process. Your manuscript titled "Blocked training facilitates learning of multiple schemas" has now been evaluated by 3 subject-matter reviewers and 1 additional code reviewer, and I include their comments at the end of this message. They find your work of interest, but raised some important points. We are interested in the possibility of publishing your study in Communications Psychology, but would like to consider your responses to these concerns and assess a revised manuscript before we make a final decision on publication.

We therefore invite you to revise and resubmit your manuscript, along with a point-by-point response to the reviewers. Please highlight all changes in the manuscript text file.

Editorially, we would like to see you provide better justification for your decision to manually turn off inference for unpredictable transitions, as well as your decision to only apply the model to explain test phase performance, while seemingly ignoring rich learning dynamics during the training phase. Please also revise your model comparison approach to mitigate potential overfitting. There is also the need to better explain how prediction errors are encoded in the model, as well as to provide better contextualization of your paradigm with respect to previous work on blocked vs. interleaved training.

Please note that your revised manuscript must comply with our formatting and reporting requirements, which are summarized on the following checklist: Communications Psychology formatting checklist and also in our style and formatting guide Communications Psychology formatting guide.

Please use the following link to submit your revised manuscript, point-by-point response to the referees' comments (which should be in a separate document to any cover letter) and the completed checklist:

[link redacted]

Please do not hesitate to contact me if you have any questions or would like to discuss these revisions further. We look forward to seeing the revised manuscript and thank you for the opportunity to review your work.

Best regards,

Jesse Rissman

Jesse Rissman, PhD
Editorial Board Member
Communications Psychology
orcid.org/0000-0001-8889-5539

EDITORIAL POLICIES AND FORMATTING

Editorial Policy: Policy requirements (Download the link to your computer as a PDF.)

* **CODE AVAILABILITY:** All Communications Psychology manuscripts must include a section titled "Code Availability" at the end of the methods section. In the event of publication, we require that the custom analysis code supporting your conclusions is made available in a publicly accessible repository; at publication, we ask you to choose a repository that provides a DOI for the code; the link to the repository and the DOI will need to be included in the Code Availability statement. Publication as Supplementary Information will not suffice. We ask you to prepare code at this stage, to avoid delays later on in the process.

* **DATA AVAILABILITY:**

All Communications Psychology manuscripts must include a section titled "Data Availability" at the end of the Methods section or main text (if no Methods). More information on this policy, is available at <http://www.nature.com/authors/policies/data/data-availability-statements-data-citations.pdf>.

At a minimum the Data availability statement must explain how the data can be obtained and whether there are any restrictions on data sharing. Communications Psychology strongly endorses open sharing of data. If you do make your data openly available, please include in the statement:

We recommend submitting the data to discipline-specific, community-recognized repositories, where possible and a list of recommended repositories is provided at

<http://www.nature.com/sdata/policies/repositories>.

If a community resource is unavailable, data can be submitted to generalist repositories such as figshare or Dryad Digital Repository. Please provide a unique identifier for the data (for example a DOI or a permanent URL) in the data availability statement, if possible. If the repository does not provide identifiers, we encourage authors to supply the search terms that will return the data. For data that have been obtained from publicly available sources, please provide a URL and the specific data product name in the data availability statement. Data with a DOI should be further cited in the methods reference section.

REVIEWERS' EXPERTISE:

Reviewer #1: context-dependent learning

Reviewer #2: computational modeling of learning and decision making

Reviewer #3: context-dependent learning

Reviewer #4: code reviewer; computational modeling of learning

REVIEWERS' COMMENTS:

Reviewer #1 (Remarks to the Author):

In this article, Beukers et al. explore the role that different learning curricula play in shaping schema composition through experiments and modeling. Experimentally, they find that blocked training leads to better performance than interleaved training, consistent with some previous results in the literature. Interestingly, they find that performance is worsened if blocked training is preceded by interleaved training, suggesting that inappropriate “splitting” during interleaved training may lead to suboptimal schema inferences that make subsequent learning difficult. The data is supported by simulations of both Bayesian and neural network models.

Comments:

The variant of the Bayesian model where inference is manually turned off for unpredictable transitions seems quite unprincipled (the authors themselves acknowledge that they do not explain this) and makes an otherwise normative model seem like a heuristic. This model variant seems unsatisfying as an explanation of the data for this reason. Can a more principled model be developed

to explain the data?

By and large, the Bayesian model is used to explain the mean performance in the test phase, but the relatively rich dynamics of learning in the training phase (in the blocked condition) are ignored. Are there specific features of the time course of learning in the training phase in the blocked condition (e.g. drops in schema performance after learning another schema; the rate of relearning a schema following schema re-exposure) that the model can or cannot explain?

Simulation 3 of the Bayesian model implies that individual differences may be able to explain the data better. Can this be explored further? Can the parameters of the model be fit to individual subjects instead of the mean behavior? Can distinct groups of subjects be found who employ different strategies?

The model simulations only show prediction accuracy. To help a reader understand the behavior of the model, would it be helpful to also plot model inferences (e.g. the time series of inferred number of contexts, context probabilities, transition probability matrices)? The authors state that in the interleaved condition, the model ends up learning schemas that do not align properly with the generative structure of the environment, but it is hard for the reader to understand what this means exactly or why it happens as no internal model variables are shown or analyzed.

The Bayesian optimal predictive distribution is approximated by 'the more psychologically plausible implementation' where only the currently active schema makes predictions. Can a formal model comparison of these two approaches be performed?

One interesting feature of the experiment is that in the blocked condition, the transition probabilities between latent schemas (not observed states) are different in the test phase than in the training phase. I would have expected this to have led to reduced performance, given that in the blocked training phase the model will have come to expect schemas to switch slowly (and it also has a stickiness bias), but in the blocked test phase the schemas switch rapidly. This reduced performance does seem to happen in the Fig. 7 model variant, but not in the Fig. 8 or Fig. 9 model variants. Can the authors comment on this? Does the Bayesian model infer new schemas in the test phase because of the change in transition probabilities?

The authors present the experiments as having 2 schemas/Markov chains with probabilistic transitions into states 3 and 4. Presumably, the experiments could also be interpreted as 4 schemas/Markov chains with deterministic transitions? Consistent with this, the motivation for turning off inference for unpredictable transitions seems to be that one of the experimentally-defined schemas is split into 2 by the model. I assume that whether or not the Bayesian model infers 1, 2, 3, 4... schemas depends on the model parameters. Is there evidence in the data that participants infer 2 rather than 4 schemas, and what would this imply about their assumptions about the generative structure of the environment?

In the test period for the audiovisual experiments, 2 movies were presented from each chain (i.e. A A B B A A ...). This is in contrast to the text experiments, where (in expectation) 1 story was presented from each chain (i.e. A B A ...), as each story was presented with 0.5 probability. Was there a reason for this change?

Reviewer #2 (Remarks to the Author):

Thank you for the opportunity to review this manuscript by Beukers and colleagues, which reports the results of a very interesting series of experiments and model simulations regarding the effect of blocked versus interleaved instruction on schema learning. Overall I found the manuscript impressive for its detailed, considerate approach to the research question, and for the organic and rigorous manner in which it married the analysis of empirical data with the interpretation of computational modelling results. I do not have any critical concerns which I feel should necessarily preclude publication of the manuscript, but I do have some questions and comments that I feel would be useful to address in a revision.

1. Better contextualization of theoretical paradigm with respect to previous work on blocked vs. interleaved training. In reading the introduction I had the distinct feeling that I was reading only one narrow perspective on a question which has been addressed more fully elsewhere. I appreciated the efforts that the manuscript went to in its Discussion (p. 27) to discuss previous research on blocked versus interleaved training, but the large volume of research referred to here (going back as far as the 1950s in some cases) only reinforced my feeling that I would have preferred to see a more thorough review of the existing literature on blocked versus interleaved learning in the *introduction* of the manuscript. Ideally this review would serve to identify the open questions which this large body of previous research has so far failed to resolve. This would be very useful for those readers who (like me) who do not have existing domain expertise in this area.

2. In Simulation 3, the manuscript handled individual differences by assuming participant-level heterogeneity in the model's concentration parameter. This is reasonable, but it is not the only individual difference that could be considered. How does a model that allows for individual differences in alpha compare with a model that allows for individual differences in lambda, or one that allows for individual differences in beta? In order to claim that individual differences in alpha are a good explanation for the heterogeneity in behaviour, it seems to me important to show (ideally via model comparison) that this varying-alpha model accounts for data better than a varying-beta or a varying-lambda model. A passing reference is made to this possibility on page 23, but in my opinion it should be treated more systematically here.

3. A related question: the models are evaluated on the basis of their MSE to human data. Unless I am mistaken, this amounts to comparing models on the basis of their goodness-of-fit without accounting for their parsimony. This is important because the best-fitting model, which allows for a different alpha value for each participant, is necessarily far less parsimonious than a model that assumes the same alpha value for all participants. Why not employ a model comparison statistic that considers both parsimony and goodness-of-fit, which would be a standard way of approaching this question? And, given that the manuscript has not done this in its current form, how can the authors be confident that Simulation 3 is not over-fitting to data?

Reviewer #3 (Remarks to the Author):

This paper addresses the problem of schema learning. Different schemas may share many of the same features but have different temporal structures. As a result, learning architectures that use a shared set of weights to represent different schemas are susceptible to catastrophic interference

(i.e., overwriting of old knowledge). The authors show through simulation that this occurs in a long short-term memory (LSTM) model. The typical fix for catastrophic interference is to interleave examples from different schemas during learning. Alternatively, “splitting” models represent knowledge about different schemas with separate sets of weights, which can prevent catastrophic interference even under blocked training. The authors conducted multiple experiments using tasks in which participants learned to predict the next event in stories that were presented via text or animated videos. The event sequences were drawn from Markov chains that represented two distinct schemas. Across the board, learning was enhanced under blocked compared to interleaved training. A Bayesian model was presented that accounts for this finding using representational splitting in response to large prediction errors. Some unique predictions of this model regarding mixed blocked/interleaved curricula were confirmed in additional experiments.

Overall, I found this research to be well-motivated and very interesting. The paper was a pleasure to read, and the results were clearly and effectively presented. I commend the authors for their commitment to replicating their findings, presenting results with/without participant exclusions, and making their materials openly available. I think this work makes a significant contribution to the literature.

Below are some comments/suggestions (in no particular order) that I hope the authors will find helpful:

1. In the Intro, the notion of “splitting” in response to large prediction errors (PEs) was a recurring theme. Yet, it was not immediately clear to me how PEs are encoded in their model. I think it would be helpful to include a description (and possibly a figure) of how the model implements splitting in response to PEs and what this looks like in blocked vs. interleaved curricula (esp. the “poor splitting” that can occur in interleaved environments).

2. I suggest moving the definition of “schema” on p. 6 to the very beginning, for those less familiar with the terminology.

3. Why use perfectly alternated sequences (ABABAB...) instead of randomly interleaved sequences during training? Do the authors think that participants could have picked up on the alternating pattern, and if so, could this have affected their behavior?

4. The colors in the Figure 2 caption don’t match the figure. Yellow (not red) should be perfectly interleaved training, and red (not purple) should be random-curriculum test phase.

5. If I understood correctly, the LSTM model outputs a probability distribution over 9 possible chain states (bottom p. 8). Can the authors clarify what the softmax probabilities in Figure 2 represent? Are these the normalized predicted probabilities for the two states that differed for chain A and B?

6. P. 11, starting with “Participants read 200 such stories...”: Please include the number of participants.

7. Figure 6 caption: I do not see (a) and (b) panels.

8. P. 17: Assigning a much larger probability to the “wrong schema” state than to each of the “wrong time step” states was not only true of the interleaved condition. It appeared to be true also for the

blocked condition.

9. P. 20: Switching off the schema inference process at the transition into states 3 and 4 does seem a bit ad hoc. In the Discussion, the authors acknowledge that future work could add a learning mechanism to the model that estimates predictability and factors the estimates into the schema inference process. Could the authors discuss what that might look like?

10. In the Intro, the authors state, “In the interleaved condition, the occurrence of prediction errors does not cleanly align with shifts in the generative model; as a result, the model ends up learning schemas that do not align properly with the generative structure of the environment” (p. 5). This raises two questions:

- a. The results from Simulation 3 suggest that high values of the model’s concentration parameter result in perfect test accuracy in the interleaved condition (Fig. 9c). Higher values of this parameter result in more splitting / more latent causes. Can the authors elaborate on this and how it relates to the above statement regarding the alignment of learned schemas with the environment?
- b. Is there a way to “reconstruct” the schemas that are learned? If so, it would be interesting to compare the learned and actual schemas for various levels of the concentration parameter.

11. Figure 10: I recommend using the same colors as in previous figures for consistency (yellow instead of purple for perfect interleaving).

Reviewer #4 (Remarks to the Author):

The authors provide an interesting and novel contribution to understanding how learning schedule affects schema acquisition from a representation-centric viewpoint, emphasizing the role of prediction errors. The paper is also accompanied by a public repository that replicates most of the results discussed in the paper. Below are my comments that examines some aspects of the code.

1. The code for reproducing figures 1 and 3 are not provided, although the authors acknowledge this in their GitHub Readme file.

2. The two betas in the code that refer to within, between, or both (e.g., line 117-124 in model.py; (code below)

```
"""
```

```
def get_beta_mode(self):  
    if self.tstep==0:  
        return 1 # between only  
    elif self.beta2_flag:  
        return 2 # between+within  
    else:  
        return 0 # within only  
    return None  
"""
```

are not explained in the manuscript, although this seems to be an interesting aspect of the model to be discussed.

3. Regarding simulations 1-4, the following function(from utils.py lines 132-)

```
"""  
def run_batch_exp(ns,args, concentration_info = None,  
stickiness_info = None,  
sparsity_info = None):  
"""
```

seems to be the core function that runs the simulation. Although this function has no problem reproducing the results of the paper, which takes into account the individual differences only within the concentration parameter, this function could be extended to incorporate potential individual differences in other parameters such as stickiness or sparsity. The current version of the function does not provide this since the lower, upper, mu, sigma variables are identical for all three parameters – this could be solved by assigning them to different variables: lower_c ,upper_c for example. Also, this function at its current state does not seem to handle cases where individual differences in the concentration parameter are not defined (i.e., no code handling exceptions where concentration_info is None). Although these points are minor since they are sufficient to reproduce the figures in the current version of the paper, these could be improved for generalizability in the future.

To the reviewers:

We are very grateful to you for your constructive feedback on our manuscript. As described in the point-by-point response below, we believe that our revised manuscript fully addresses the comments raised during the prior round of reviews. Note that we have also marked substantive changes within the manuscript file itself using blue text.

Best wishes,
Ken Norman (on behalf of the full set of authors)

REVIEWERS' COMMENTS:

Reviewer #1 (Remarks to the Author):

In this article, Beukers et al. explore the role that different learning curricula play in shaping schema composition through experiments and modeling. Experimentally, they find that blocked training leads to better performance than interleaved training, consistent with some previous results in the literature. Interestingly, they find that performance is worsened if blocked training is preceded by interleaved training, suggesting that inappropriate “splitting” during interleaved training may lead to suboptimal schema inferences that make subsequent learning difficult. The data is supported by simulations of both Bayesian and neural network models.

Comments:

The variant of the Bayesian model where inference is manually turned off for unpredictable transitions seems quite unprincipled (the authors themselves acknowledge that they do not explain this) and makes an otherwise normative model seem like a heuristic. This model variant seems unsatisfying as an explanation of the data for this reason. Can a more principled model be developed to explain the data?

We thank the reviewer for raising this issue, which we think is important to address in future work. We now discuss this direction on page 31:

“Also, the Bayesian model currently assumes that participants can ignore unpredictable transitions when making inferences about latent causes, but it does not actually explain this. Future work could add a learning mechanism to the model that estimates predictability and factors these predictability estimates into inferences, as hypothesized by normative models of learning (Mathys et al., 2014; Piray & Daw, 2021). The basic idea is that learning should only occur for predictable (i.e., learnable) transitions. Because predictability is a latent variable, a more sophisticated model would attenuate updating in proportion to inferred unpredictability.”

By and large, the Bayesian model is used to explain the mean performance in the test phase, but the relatively rich dynamics of learning in the training phase (in the blocked condition) are ignored. Are there specific features of the time course of learning in the training phase in the

blocked condition (e.g. drops in schema performance after learning another schema; the rate of relearning a schema following schema re-exposure) that the model can or cannot explain?

In the previous version of the paper, we showed the model's predictions about training accuracy in all of the model figures, but we agree with the reviewer that we could have done more to discuss the model's ability to explain particular features of the training-phase data. We have added Figure S5, which overlays the Simulation 2 fit on the blocked and interleaved data from the original experiment.

Figure S5: Model fits for Simulation 2, overlaid on human data from the original blocked and interleaved experiments.

We have also added the following text on page 21:

“Looking at the full trajectory of training accuracy across trials, the model succeeds in capturing the large dip in performance at the first block boundary in the blocked condition. There were, however, a few minor discrepancies between the model fits and human performance: The model predicts a somewhat steeper rise in performance in the first block than is apparent in human data. Also, the model slightly underestimates the size of the dip in performance at the first block boundary, and it does not capture the smaller dips in performance that are evident in the human data at the second and third block boundaries and at the start of the test phase (to facilitate these comparisons, Figure S5 shows the best model fit and the human data overlaid on the same plot). Speculatively, these small discrepancies can be attributed to human participants being inattentive (e.g., failing to notice a shift from the Cafe to the Brew House) whereas the model does not suffer from attentional lapses.”

Simulation 3 of the Bayesian model implies that individual differences may be able to explain the data better. Can this be explored further? Can the parameters of the model be fit to individual subjects instead of the mean behavior? Can distinct groups of subjects be found who employ different strategies?

We think that fitting the model to individual-subject data is a promising direction. Unfortunately, though, this is not feasible using our current model fitting procedure, which involves running

size-2000 job arrays (each of which runs for 6 hours) on our cluster. Running this for each participant in our studies would take more time than we were allotted for the revision. Furthermore, showing (in a non-circular way) that individual differences in model parameters are responsible for differences in learning would require us to (first) estimate these parameters using a separate task battery for each participant, and then relate these (separately estimated) parameters to parameters estimated from the schema learning task. Developing a task battery that yields reliable individual-difference estimates of concentration, stickiness, and sparsity is a key goal for our future research (one we are actively working on; see Zorowitz & Niv, *Biological Psychiatry Cognitive Neuroscience and Neuroimaging*, 2023, for discussion of the challenges inherent achieving good reliability), but we are not there yet. In the Discussion section (page 30), we emphasize the importance of this individual-differences work as a future direction for our research:

“Another future direction is to improve our explanation of individual differences: Why is it that some participants seem to learn in the interleaved condition but others do not? As mentioned earlier, one promising approach would be to estimate individual differences in key parameters governing latent cause inference (e.g., the concentration parameter, α), using a separate set of tasks; we could then see whether these estimated parameters relate to interleaved learning performance in a way that aligns with the predictions of our Bayesian model (for other work exploring individual differences in latent cause parameters, see, e.g., Gershman and Hartley, 2015; Norbury et al., 2022; Zika et al., 2022).”

The model simulations only show prediction accuracy. To help a reader understand the behavior of the model, would it be helpful to also plot model inferences (e.g. the time series of inferred number of contexts, context probabilities, transition probability matrices)? The authors state that in the interleaved condition, the model ends up learning schemas that do not align properly with the generative structure of the environment, but it is hard for the reader to understand what this means exactly or why it happens as no internal model variables are shown or analyzed.

Thanks for this suggestion. The previous version of the manuscript did report the number of inferred latent causes in Simulation 3 as a function of the concentration parameter (Figure 9e) but there is more that we could have done in this regard. To address this, we have added a new Figure S6 that includes (part a) a histogram of the inferred number of latent causes in each of the model conditions for the model used in Simulation 3 and (part b) plots of the learned transition matrices in Simulation 3 in the case where the model infers two latent causes (in either the interleaved or blocked condition), and (part c) a plot of the learned transition matrix in Simulation 3 in the case where the model only infers one latent cause (in the interleaved condition). This latter plot illustrates that, when the model only infers one latent cause, it learns a transition matrix that averages across the ground-truth transition matrices for the two Markov chains.

Figure S6: Analysis of the number of inferred latent causes and their properties. (a) Histogram of the number of inferred latent causes in each of the model conditions for the model used in Simulation 3. (b) Plot of the learned transition matrices in Simulation 3 in the case where the model infers two latent causes (in either the interleaved or blocked condition), and (c) plot of the learned transition matrix in Simulation 3 in the case where the model only infers one latent cause (in the interleaved condition). When the model infers two latent causes, the learned transition matrices accurately reflect the ground truth transition structure; when the model only infers one latent cause, it learns a transition matrix that averages across the ground-truth transition matrices for the two Markov chains.

We have also added the following text on pages 23-24:

“Figure S6 shows a histogram of the number of latent causes inferred in the different conditions of the experiment, confirming that the model mostly (though not always) succeeded in inferring two latent causes in the blocked condition, whereas it was less prone to find this two-cause solution in the interleaved condition; the figure also shows that, when the model did find a two-cause solution, the learned transition matrices matched the ground-truth transition structure, whereas when it found a one-cause solution (in the interleaved condition) the learned transition matrix averaged across the two ground-truth structures (and thus did not accurately approximate either of them).”

The Bayesian optimal predictive distribution is approximated by ‘the more psychologically plausible implementation’ where only the currently active schema makes predictions. Can a formal model comparison of these two approaches be performed?

We now include this comparison. The result is, essentially, that the two models are equally good at fitting the data. The model fits for the Bayesian-optimal model are now included in Figure S9.

Figure S9: Results from a variant of Simulation 3 (compare to Figures 9 and 10) where we use the Bayesian optimal predictive distribution (marginalizing across schemas) rather than our default approximation where only the currently active schema is used to make predictions. (a,b) Model accuracy over time on blocked and interleaved curricula, respectively. (c) Violin plot of model test accuracy. (d-f) Increased concentration leads the model to split more regularly in the interleaved condition, which improves performance on average. Each dot represents a model run using a slightly different concentration parameter. (d) Number of latent causes versus test accuracy; (e) Concentration versus number of latent causes; (f) Concentration versus test accuracy. (g-i) Model predictions on early, middle and late curricula, respectively. (j) Violin plots of test accuracy predicted by the model on each curriculum. Best-fit parameters were obtained using a new gridsearch for this model variant. Overall, model fits are qualitatively similar to those in Simulation 3, although the fit to human data is quantitatively worse when we use the Bayesian optimal predictive distribution (depicted here) vs. when only the currently active schema is used to make predictions (in Simulation 3). Here, $MSE=0.0432$, compared to $MSE=0.0371$ in Simulation 3. This MSE difference corresponds to an AIC difference of 122.12 (AIC for Bayesian optimal approach = -2507.87; AIC for approximation = -2629.96).

We have also made the following edits to the text on page 44:

“Marginalizing across schemas can be computationally (and thus psychologically) cumbersome, especially as the number of inferred schemas increases. This led us to default to the less-cumbersome (and thus more-psychologically-plausible) approximation wherein only the currently active schema is used to make predictions (Anderson, 1991)…”

Note that, for completeness, we also ran a variant of Simulation 3 using the Bayesian optimal approach. The results of this simulation are shown in Figure S9; overall, the fits provided by the Bayesian-optimal approach were qualitatively similar but quantitatively slightly less good.”

One interesting feature of the experiment is that in the blocked condition, the transition probabilities between latent schemas (not observed states) are different in the test phase than in the training phase. I would have expected this to have led to reduced performance, given that in the blocked training phase the model will have come to expect schemas to switch slowly (and it also has a stickiness bias), but in the blocked test phase the schemas switch rapidly. This reduced performance does seem to happen in the Fig. 7 model variant, but not in the Fig. 8 or Fig. 9 model variants. Can the authors comment on this? Does the Bayesian model infer new schemas in the test phase because of the change in transition probabilities?

As a clarification, in Figure 7, performance is not specifically reduced in the final test – it is also reduced in blocks 2 and 4 during training, reflecting the fact that the schema is not learned properly in the first place. To answer the reviewer’s question about whether the change in transition probabilities in the test phase causes the model to infer new schemas: The answer to this question is no – the model only learns transition probabilities between *observable* states and does not have any way of learning transition probabilities between *latent* states. Because the model has not learned transition probabilities between latent states, it is not surprised when these transition probabilities change at the start of the test phase (and thus it does not infer new schemas at this point). In principle, one could build a hierarchical model that learns transitions between latent states in addition to learning transitions between observable states, but that would be a different (and substantially more complex) model – the success of our model at explaining the data suggests that learning transition probabilities between latent states is not required in this particular case.

The authors present the experiments as having 2 schemas/Markov chains with probabilistic transitions into states 3 and 4. Presumably, the experiments could also be interpreted as 4 schemas/Markov chains with deterministic transitions? Consistent with this, the motivation for turning off inference for unpredictable transitions seems to be that one of the experimentally-defined schemas is split into 2 by the model. I assume that whether or not the Bayesian model infers 1, 2, 3, 4... schemas depends on the model parameters. Is there evidence in the data that participants infer 2 rather than 4 schemas, and what would this imply about their assumptions about the generative structure of the environment?

Given the way that the task was defined, the behavioral predictions that a participant would make if they had two vs. four schemas would be identical, so unfortunately there is no way to discriminate between these two alternatives based on the data from this study. Another approach to this question would be to look at neural data (e.g., using representational similarity analysis) to assess whether participants’ neural representations are consistent with the two-schema and four-schema solutions, respectively. In the revised manuscript, we have added the following passage to the Discussion on pages 31-32 to alert the reader to the possibility of the

“four-schema” solution and the potential usefulness of neural data in identifying whether participants are representing the task in this fashion:

“Finally, another future direction is to use fMRI and other neural measures to seek converging evidence for the claims made here. While our model predicts that participants will (mostly) learn a two-schema solution in the blocked condition, other solutions are possible; in particular, participants could learn a four-schema solution where each of the four “paths” in Figure 1 is assigned its own schema. Behavioral prediction-accuracy data can not, on its own, distinguish between the two-schema and four-schema possibilities (since they both support accurate prediction). However, fMRI data could potentially be used to tease them apart – e.g., one could use representational similarity analysis (Kriegeskorte et al., 2008) to determine whether there is a pattern that is shared by all states that are part of chain A and not by states part of chain B (indicative of a two-schema solution), and/or whether there is a pattern that is shared by all states on the left “path” of chain A but not the right “path” of chain A (indicative of a four-schema solution).”

In the test period for the audiovisual experiments, 2 movies were presented from each chain (i.e. A A B B A A ...). This is in contrast to the text experiments, where (in expectation) 1 story was presented from each chain (i.e. A B A ...), as each story was presented with 0.5 probability. Was there a reason for this change?

This A A B B A A ... structure was put in place for reasons unrelated to the work presented here. We now include a brief note to this effect on page 38:

“...this alternating structure was put in place for reasons unrelated to the work presented here (the audiovisual experiments included additional episodic memory tests at the very end of the experiment session – for these episodic memory analyses, we wanted to make sure that we had a roughly equal number of temporally-adjacent “same-schema” and “different-schema” weddings; data from these tests will be described in another paper).”

Reviewer #2 (Remarks to the Author):

Thank you for the opportunity to review this manuscript by Beukers and colleagues, which reports the results of a very interesting series of experiments and model simulations regarding the effect of blocked versus interleaved instruction on schema learning. Overall I found the manuscript impressive for its detailed, considerate approach to the research question, and for the organic and rigorous manner in which it married the analysis of empirical data with the interpretation of computational modelling results. I do not have any critical concerns which I feel should necessarily preclude publication of the manuscript, but I do have some questions and comments that I feel would be useful to address in a revision.

1. Better contextualization of theoretical paradigm with respect to previous work on blocked vs. interleaved training. In reading the introduction I had the distinct feeling that I was reading only one narrow perspective on a question which has been addressed more fully elsewhere. I

appreciated the efforts that the manuscript went to in its Discussion (p. 27) to discuss previous research on blocked versus interleaved training, but the large volume of research referred to here (going back as far as the 1950s in some cases) only reinforced my feeling that I would have preferred to see a more thorough review of the existing literature on blocked versus interleaved learning in the *introduction* of the manuscript. Ideally this review would serve to identify the open questions which this large body of previous research has so far failed to resolve. This would be very useful for those readers who (like me) who do not have existing domain expertise in this area.

Thank you for this suggestion. We did not want to overload the reader with a detailed literature review of blocked vs. interleaved learning in the Introduction, especially as much of it is not directly relevant to the present study (e.g., the work from Carvalho and Goldstone on how blocked vs. interleaved study affects how participants attend to different dimensions of complex visual stimuli – as noted in the Discussion, this is not relevant to our study, where the features defining the individual states in the stories were very obvious; it would be distracting to readers to mention this work in the Introduction and then explain why it is not relevant). Having said this, we agree that it would be useful to give readers a bit more of a sense in the Introduction of the “gap” in the literature that is filled by our work. We still rely mainly on the Discussion to go into details of the literature on blocked vs. interleaved learning, but we have added the following text to the Introduction on page 3:

“One way to avoid CI is to interleave new experiences with old knowledge during learning (McClelland, 2013). By continually pressuring the network weights to maintain old knowledge while encoding new information, interleaved learning allows the weights to settle into a state that jointly represents new and old information. *In keeping with these simulation results, numerous studies have shown benefits of interleaved learning (e.g., Kornell and Bjork, 2008; Schmidt and Bjork, 1992; Yan et al., 2017).*”

and on pages 4-5:

“A key prediction of splitting models is that interleaving is not necessary to avoid catastrophic interference: If the model is given a block of training on inputs that follow one schema, and then switches to training on inputs that follow a new schema, this un signaled transition between schemas should trigger a large prediction error. That, in turn, will cause the model to split off the representation of the new schema from the old schema, preventing CI. By contrast, models without the capacity for splitting predict that CI will happen in this blocked-learning situation -- i.e. during the second block, the schema learned in the first block would be overwritten. *Note that prior studies that have manipulated blocked vs. interleaved learning have not examined learning of multiple schemas (where schema transitions are not marked and the features of the schemas are unknown), so this prediction remains to be tested.*”

2. In Simulation 3, the manuscript handled individual differences by assuming participant-level heterogeneity in the model's concentration parameter. This is reasonable, but it is not the only individual difference that could be considered. How does a model that allows for individual

differences in alpha compare with a model that allows for individual differences in lambda, or one that allows for individual differences in beta? In order to claim that individual differences in alpha are a good explanation for the heterogeneity in behaviour, it seems to me important to show (ideally via model comparison) that this varying-alpha model accounts for data better than a varying-beta or a varying-lambda model. A passing reference is made to this possibility on page 23, but in my opinion it should be treated more systematically here.

Thanks for this suggestion. We ran these simulations and found that varying-alpha, varying-beta, and varying-lambda all account for the data equally well. The results for varying-beta and varying-lambda are now included in Figures S7 and S8, respectively. The take-away point from these new simulations is that variance in any parameter that affects splitting (i.e., all three of the aforementioned parameters) can give rise to the bimodal distribution of performance that we observed in the interleaved condition. We have adjusted the text on page 24 accordingly:

“In other simulations, we found that allowing the stickiness and sparsity parameters to vary across model runs also leads to good overall model fits (Figure S7 shows the results for varying-stickiness simulations, and Figure S8 shows the results for varying-sparsity simulations; quantitatively the MSE was similar across conditions: 0.0376 for varying-sparsity and 0.0377 for varying-stickiness, compared to 0.0371 for varying-concentration). These results show that there is nothing special about the concentration parameter per se -- all three of the main model parameters affect whether splitting occurs when the generative model shifts in the interleaved condition, and consequently variance in any of these parameters can affect whether learning succeeds or fails in the interleaved condition.”

Figure S7: Individual differences in model stickiness can potentially explain human performance in the interleaved condition (compare to Figure 9). (a,b) Model accuracy over time on blocked and interleaved curricula, respectively. (c) Violin plot of model test accuracy. (d-f) Reduced stickiness leads the model to split more regularly in the interleaved condition, which improves performance. Each dot represents a model run using a slightly different concentration

parameter. (d) Number of latent causes versus test accuracy; (e) Stickiness versus number of latent causes; (f) Stickiness versus test accuracy.

“Figure S8: Individual differences in model sparsity can potentially explain human performance in the interleaved condition (compare to Figure 9). (a,b) Model accuracy over time on blocked and interleaved curricula, respectively. (c) Violin plot of model test accuracy. (d-f) Reduced sparsity leads the model to split more regularly in the interleaved condition, which improves performance. Each dot represents a model run using a slightly different concentration parameter. (d) Number of latent causes versus test accuracy; (e) Sparsity versus number of latent causes; (f) Sparsity versus test accuracy.”

3. A related question: the models are evaluated on the basis of their MSE to human data. Unless I am mistaken, this amounts to comparing models on the basis of their goodness-of-fit without accounting for their parsimony. This is important because the best-fitting model, which allows for a different alpha value for each participant, is necessarily far less parsimonious than a model that assumes the same alpha value for all participants. Why not employ a model comparison statistic that considers both parsimony and goodness-of-fit, which would be a standard way of approaching this question? And, given that the manuscript has not done this in its current form, how can the authors be confident that Simulation 3 is not over-fitting to data?

We appreciate this suggestion, but we don't think that it's applicable here, for multiple reasons.

First, all of the models have the same number of free parameters for model-fitting purposes. In Simulation 3, as in Simulation 2, we are fitting the model to the average data across participants, not individual participants. In Simulation 3, concentration is randomly sampled from a Gaussian of fixed, nonzero variance for each simulated participant; in Simulation 2, concentration is set to the mean of that Gaussian, which is equivalent to sampling from a

Gaussian with zero variance. Since the variance is fixed in both cases (i.e., only the mean concentration value is adjusted during the model-fitting process), the number of free parameters is the same.

Second, the reason why we prefer Simulation 3 over Simulation 2 is not because of a difference in MSE; rather, it's because Simulation 2 (which relies on random variance in the sequence of trials to explain variance in learning outcomes in the blocked condition) makes an incorrect prediction about which sequences of trials will lead to the best performance, as described in the paper. We have added the following text on page 22 to emphasize this point:

“As described above, Simulation 2 relied entirely on sequence variability to explain variability in human performance; this approach led to incorrect predictions about variability across sequences, so we reject this approach despite it leading to a good fit to the average human performance curve (measured in terms of mean squared error).”

Reviewer #3 (Remarks to the Author):

This paper addresses the problem of schema learning. Different schemas may share many of the same features but have different temporal structures. As a result, learning architectures that use a shared set of weights to represent different schemas are susceptible to catastrophic interference (i.e., overwriting of old knowledge). The authors show through simulation that this occurs in a long short-term memory (LSTM) model. The typical fix for catastrophic interference is to interleave examples from different schemas during learning. Alternatively, “splitting” models represent knowledge about different schemas with separate sets of weights, which can prevent catastrophic interference even under blocked training. The authors conducted multiple experiments using tasks in which participants learned to predict the next event in stories that were presented via text or animated videos. The event sequences were drawn from Markov chains that represented two distinct schemas. Across the board, learning was enhanced under blocked compared to interleaved training. A Bayesian model was presented that accounts for this finding using representational splitting in response to large prediction errors. Some unique predictions of this model regarding mixed blocked/interleaved curricula were confirmed in additional experiments.

Overall, I found this research to be well-motivated and very interesting. The paper was a pleasure to read, and the results were clearly and effectively presented. I commend the authors for their commitment to replicating their findings, presenting results with/without participant exclusions, and making their materials openly available. I think this work makes a significant contribution to the literature.

Below are some comments/suggestions (in no particular order) that I hope the authors will find helpful:

1. In the Intro, the notion of “splitting” in response to large prediction errors (PEs) was a recurring theme. Yet, it was not immediately clear to me how PEs are encoded in their model. I

think it would be helpful to include a description (and possibly a figure) of how the model implements splitting in response to PEs and what this looks like in blocked vs. interleaved curricula (esp. the “poor splitting” that can occur in interleaved environments).

Thank you for this suggestion. We have edited the text to the Introduction to address this point (pages 17-18):

“On each timestep, the model observes a state and attempts to predict the following state. To do this, the model selects the most probable latent cause and uses the empirically estimated state transition matrix for that latent cause to calculate the probability of each possible transition. After issuing a prediction and observing the environment transition, the model then updates the posterior probability of each cause in its library, where this posterior probability is a function of the likelihood of the observed transition under each cause and its prior probability; the model also estimates the probability that a new latent cause (not previously used) is present.

Intuitively, the model's prediction error is inversely proportional to the likelihood of the observed transition under the currently-active latent cause. When a large prediction error is observed, that prediction error makes the current latent cause improbable; this, in turn, increases the odds that the most probable latent cause will be a different, previously-used latent cause (in which case the model will switch to that cause) or the new, never-before-used latent cause (in which case the model will split off a new latent cause).”

Also, to provide further insight into the model’s splitting behavior, we have added a new Figure S6 that includes (part a) a histogram of the inferred number of latent causes in each of the model conditions for the model used in Simulation 3 and (part b) plots of the learned transition matrices in Simulation 3 in the case where the model infers two latent causes (in either the interleaved or blocked condition), and (part c) a plot of the learned transition matrix in Simulation 3 in the case where the model only infers one latent cause (in the interleaved condition). This latter plot illustrates that, when the model only infers one latent cause, it learns a transition matrix that averages across the ground-truth transition matrices for the two Markov chains.

Figure S6: Analysis of the number of inferred latent causes and their properties. (a) Histogram of the number of inferred latent causes in each of the model conditions for the model used in Simulation 3. (b) Plot of the learned transition matrices in Simulation 3 in the case where the model infers two latent causes (in either the interleaved or blocked condition), and (c) plot of the learned transition matrix in Simulation 3 in the case where the model only infers one latent cause (in the interleaved condition). When the model infers two latent causes, the learned transition matrices accurately reflect the ground truth transition structure; when the model only infers one latent cause, it learns a transition matrix that averages across the ground-truth transition matrices for the two Markov chains.

We also added the following text on pages 23-24:

“Figure S6 shows a histogram of the number of latent causes inferred in the different conditions of the experiment, confirming that the model mostly (though not always) succeeded in inferring two latent causes in the blocked condition, whereas it was less prone to find this two-cause solution in the interleaved condition; the figure also shows that, when the model did find a two-cause solution, the learned transition matrices matched the ground-truth transition structure, whereas when it found a one-cause solution (in the interleaved condition) the learned transition matrix averaged across the two ground-truth structures (and thus did not accurately approximate either of them).”

2. I suggest moving the definition of “schema” on p. 6 to the very beginning, for those less familiar with the terminology.

Done. The start of the paper now reads (page 3):

“Over the course of a lifetime, we acquire schematic knowledge of how different events typically unfold; for example, we know what to expect when we go through airport security or order at a restaurant (Baldassano et al., 2017; Rumelhart, 1975; Schank and Abelson, 1977). For the

purpose of this paper, we define a schema as a learned mental model used for predicting upcoming states in the environment (e.g., Bower et al., 1979; Mandler, 1984; Schank and Abelson, 1977)."

3. Why use perfectly alternated sequences (ABABAB...) instead of randomly interleaved sequences during training? Do the authors think that participants could have picked up on the alternating pattern, and if so, could this have affected their behavior?

We did this to ensure that we did not inadvertently obtain a long run of stories from one chain during training (as this would make learning dynamics more similar across the blocked and interleaved conditions, reducing differences between these conditions). To make this more concrete, if a participant in interleaved condition were (by chance) given eight stories in a row from one chain, the resulting prediction error when the schema switched might be sufficient to trigger splitting, which would help learning. We now mention this point on page 7:

"We used this strictly alternating structure to ensure that we did not inadvertently obtain a long run of stories from one chain during training (as this would make learning dynamics more similar across the blocked and interleaved conditions, reducing differences between these conditions)."

As for whether participants could have picked up on this alternating pattern: potentially, but – if they did – it was not sufficient to reliably lead to successful learning. Consider the condition where participants were explicitly instructed to attend to the cafe / brew house information: It seems very likely that participants in this condition were aware of the strict alternation, but average test performance in this condition was still quite poor (68% accuracy).

4. The colors in the Figure 2 caption don't match the figure. Yellow (not red) should be perfectly interleaved training, and red (not purple) should be random-curriculum test phase.

Thanks for noticing this; we have fixed this error.

5. If I understood correctly, the LSTM model outputs a probability distribution over 9 possible chain states (bottom p. 8). Can the authors clarify what the softmax probabilities in Figure 2 represent? Are these the normalized predicted probabilities for the two states that differed for chain A and B?

Yes, that is correct. We now state this explicitly in the Figure 2 caption on page 10:

"... y-axis is the softmax activation, interpreted as the network's probability estimate that the next state will be the ``chain A" (green line) or ``chain B" (blue line) state that follows the current state."

and also in the text on page 9:

“Figure 2 plots the network’s softmax activations, interpreted as the network’s probability estimate that the next state will be the state that follows the current one according to chain A vs. chain B, after being given test inputs that were actually generated from chain A (left) or chain B (right) under interleaved (top) or blocked (bottom) training.”

6. P. 11, starting with “Participants read 200 such stories...”: Please include the number of participants.

We have now added a pointer to Figure 3 for the number of participants (it seemed awkward to mention it here since the number of participants varied across conditions and also across replications of the experiment). See page 11:

“Participants read 200 such stories, one sentence at a time, and continually made 2-alternative forced choice (2AFC) predictions about what will happen next in the story (see Figure 3 for numbers of participants in each condition).”

7. Figure 6 caption: I do not see (a) and (b) panels.

Fixed; before, we left off panel b!

8. P. 17: Assigning a much larger probability to the “wrong schema” state than to each of the “wrong time step” states was not only true of the interleaved condition. It appeared to be true also for the blocked condition.

We have rewritten this passage (pages 16-17) to reflect this point:

“The same pattern was present in the blocked condition: The AMT blocked participant group assigned a higher probability to the “wrong chain” state than the probability they assigned, on average, to each of the “wrong time step” states, $t(65)=8.84$, $d=1.09$, $p<0.001$. Similarly, the undergraduate blocked participant group assigned a higher probability to the “wrong chain” state than the probability they assigned, on average, to each of the “wrong time step” states, $t(14)=4.48$, $d=1.16$, $p<0.001$.”

9. P. 20: Switching off the schema inference process at the transition into states 3 and 4 does seem a bit ad hoc. In the Discussion, the authors acknowledge that future work could add a learning mechanism to the model that estimates predictability and factors the estimates into the schema inference process. Could the authors discuss what that might look like?

We thank the reviewer for raising this issue, which we think is important to address in future work. We now discuss this direction on page 31:

“Also, the Bayesian model currently assumes that participants can ignore unpredictable transitions when making inferences about latent causes, but it does not actually explain this. Future work could add a learning mechanism to the model that estimates predictability and factors these predictability estimates into inferences, as hypothesized by normative models of learning (Mathys et al., 2014; Piray & Daw, 2021). The basic idea is that learning should only occur for predictable (i.e., learnable) transitions. Because predictability is a latent variable, a more sophisticated model would attenuate updating in proportion to inferred unpredictability.”

10. In the Intro, the authors state, “In the interleaved condition, the occurrence of prediction errors does not cleanly align with shifts in the generative model; as a result, the model ends up learning schemas that do not align properly with the generative structure of the environment” (p. 5). This raises two questions:

a. The results from Simulation 3 suggest that high values of the model’s concentration parameter result in perfect test accuracy in the interleaved condition (Fig. 9c). Higher values of this parameter result in more splitting / more latent causes. Can the authors elaborate on this and how it relates to the above statement regarding the alignment of learned schemas with the environment?

We agree with the reviewer that the initial description of the simulation results in the Introduction did not fit perfectly with the actual simulation results. To address this, we have rewritten the passage on page 5 in the Introduction to better fit with the actual simulation results:

“Notably, our model accounts for poor performance in the interleaved condition in terms of failure to split. In the interleaved condition, the model does not reliably experience large prediction errors when the generative model shifts, so the model often (but not always) fails to split, instead ending up with a single schema that does not align properly with the generative structure of the environment (and thus does not support accurate prediction).”

b. Is there a way to “reconstruct” the schemas that are learned? If so, it would be interesting to compare the learned and actual schemas for various levels of the concentration parameter.

Yes, this information is now contained in Supplementary Figure S6, which shows plots of the learned transition matrices in Simulation 3 in the case where the model infers two latent causes (in either the interleaved or blocked condition), and in the case where the model only infers one latent cause (in the interleaved condition). This latter plot illustrates that, when the model only

infers one latent cause, it learns a transition matrix that averages across the ground-truth transition matrices for the two Markov chains.

Figure S6: Analysis of the number of inferred latent causes and their properties. (a) Histogram of the number of inferred latent causes in each of the model conditions for the model used in Simulation 3. (b) Plot of the learned transition matrices in Simulation 3 in the case where the model infers two latent causes (in either the interleaved or blocked condition), and (c) plot of the learned transition matrix in Simulation 3 in the case where the model only infers one latent cause (in the interleaved condition). When the model infers two latent causes, the learned transition matrices accurately reflect the ground truth transition structure; when the model only infers one latent cause, it learns a transition matrix that averages across the ground-truth transition matrices for the two Markov chains.

11. Figure 10: I recommend using the same colors as in previous figures for consistency (yellow instead of purple for perfect interleaving).

Fixed.

Reviewer #4 (Remarks to the Author):

The authors provide an interesting and novel contribution to understanding how learning schedule affects schema acquisition from a representation-centric viewpoint, emphasizing the role of prediction errors. The paper is also accompanied by a public repository that replicates most of the results discussed in the paper. Below are my comments that examines some aspects of the code.

1. The code for reproducing figures 1 and 3 are not provided, although the authors acknowledge this in their GitHub Readme file.

We now include code to reproduce all of the figures (except for Figure 1, which is an illustration).

2. The two betas in the code that refer to within, between, or both (e.g., line 117-124 in model.py; (code below)

```
"""  
def get_beta_mode(self):  
if self.tstep==0:  
return 1 # between only  
elif self.beta2_flag:  
return 2 # between+within  
else:  
return 0 # within only  
return None  
"""
```

are not explained in the manuscript, although this seems to be an interesting aspect of the model to be discussed.

This allowed for the beta (stickiness) parameter to be set differently for transitions within and across stories. We did not take advantage of this flexibility in the manuscript. We have now rewritten the code to eliminate this unused flexibility.

3. Regarding simulations 1-4, the following function (from utils.py lines 132-)

```
"""  
def run_batch_exp(ns,args, concentration_info = None,  
stickiness_info = None,  
sparsity_info = None):  
"""
```

seems to be the core function that runs the simulation. Although this function has no problem reproducing the results of the paper, which takes into account the individual differences only within the concentration parameter, this function could be extended to incorporate potential individual differences in other parameters such as stickiness or sparsity. The current version of the function does not provide this since the lower, upper, mu, sigma variables are identical for all three parameters – this could be solved by assigning them to different variables: lower_c ,upper_c for example. Also, this function at its current state does not seem to handle cases where individual differences in the concentration parameter are not defined (i.e., no code handling exceptions where concentration_info is None). Although these points are minor since they are sufficient to reproduce the figures in the current version of the paper, these could be improved for generalizability in the future.

The code now allows users to independently specify whether concentration, stickiness, and sparsity will vary across seeds. When the concentration_info, stickiness_info, or sparsity_info dictionaries are None, this will make the code not vary that parameter across seeds. In other words: if you would like to have no differences in a parameter across seeds, then you would not

pass in a parameterX_info dictionary for that parameter. But, if you do want to try running the simulations with that parameter varying across seeds, then you will pass a parameterX_info dictionary for that parameter to the simulation driver.

We have provided more documentation in the corresponding notebook (https://colab.research.google.com/github/PrincetonCompMemLab/blocked_training_facilitates_learning/blob/master/new_simulation_sandbox.ipynb) to make this clear so that users can easily try different combinations.

7th Feb 24

Dear Professor Norman,

Your manuscript titled "Blocked training facilitates learning of multiple schemas" has now been seen by our reviewers, whose comments appear below. In light of their advice I am delighted to say that we are happy, in principle, to publish a suitably revised version in Communications Psychology under the open access CC BY license (Creative Commons Attribution v4.0 International License).

We therefore invite you to revise your paper one last time to address the remaining concerns of our reviewers and a list of editorial requests. At the same time we ask that you edit your manuscript to comply with our format requirements and to maximise the accessibility and therefore the impact of your work.

EDITORIAL REQUESTS:

SUBMISSION INFORMATION:

OPEN ACCESS:

Communications Psychology is a fully open access journal. Articles are made freely accessible on publication under a CC BY license (Creative Commons Attribution 4.0 International License). This license allows maximum dissemination and re-use of open access materials and is preferred by many research funding bodies.

For further information about article processing charges, open access funding, and advice and support from Nature Research, please visit <https://www.nature.com/commspsychol/article-processing-charges>

At acceptance, you will be provided with instructions for completing this CC BY license on behalf of all authors. This grants us the necessary permissions to publish your paper. Additionally, you will be asked to declare that all required third party permissions have been obtained, and to provide billing information in order to pay the article-processing charge (APC).

* TRANSPARENT PEER REVIEW: Communications Psychology uses a transparent peer review system. On author request, confidential information and data can be removed from the published reviewer

reports and rebuttal letters prior to publication. If you are concerned about the release of confidential data, please let us know specifically what information you would like to have removed. Please note that we cannot incorporate redactions for any other reasons.

* CODE AVAILABILITY: All Communications Psychology manuscripts must include a section titled "Code Availability" at the end of the methods section. We require that the custom analysis code supporting your conclusions is made available in a publicly accessible repository at this stage; please choose a repository that generates a digital object identifier (DOI) for the code; the link to the repository and the DOI must be included in the Code Availability statement. Publication as Supplementary Information will not suffice.

* DATA AVAILABILITY:

[link redacted]

Best regards,

Marike, on behalf of Jesse Rissman

Marike Schiffer, PhD
Chief Editor
Communications Psychology

REVIEWERS' COMMENTS:

Reviewer #1 (Remarks to the Author):

I thank for the authors for performing a formal comparison with a variant of the model that uses the Bayesian optimal predictive distribution and for making a new figure showing the inferred transition probabilities.

I am satisfied with the responses and the caveats added to the text.

My only remaining question is regarding the transition probabilities shown in Figure S6. For both chain A and chain B, there seems to be zero probability mass at the transitions where the chains branch and merge (e.g. for chain A, transitioning from state 1 to state 3 or 4 (branching), and transitioning from state 7 or 8 to state 9 (merging)). Can the authors comment on this apparent discrepancy between the true and inferred transition probabilities. It might be helpful to also add a colorbar indicating the mapping from probabilities to colors.

Reviewer #2 (Remarks to the Author):

I am satisfied with the authors' responses to my previous comments.

Reviewer #3 (Remarks to the Author):

The authors adequately addressed all of my concerns.

Reviewer #4 (Remarks to the Author):

The authors have addressed my previous concerns.

Response to Reviewer 1

Reviewer #1 (Remarks to the Author):

I thank for the authors for performing a formal comparison with a variant of the model that uses the Bayesian optimal predictive distribution and for making a new figure showing the inferred transition probabilities.

I am satisfied with the responses and the caveats added to the text.

My only remaining question is regarding the transition probabilities shown in Figure S6. For both chain A and chain B, there seems to be zero probability mass at the transitions where the chains branch and merge (e.g. for chain A, transitioning from state 1 to state 3 or 4 (branching), and transitioning from state 7 or 8 to state 9 (merging)). Can the authors comment on this apparent discrepancy between the true and inferred transition probabilities. It might be helpful to also add a colorbar indicating the mapping from probabilities to colors.

Thanks for mentioning this. Regarding the “branching” transition from states 1 and 2 to the fully-unpredictable states (3 and 4): This unpredictable transition was not shown to this version of the model (see discussion of this point in Simulation 2, and also in the Limitations section of the Discussion). Regarding the “merging” transition from states 7 and 8 to state 9: State 9 was not shown to the model since it occurs in identical form at the end of every story and participants in our experiments were never asked to predict it (in the text narrative experiments, State 9 corresponded to the statement, “That is all that is remembered”); we should emphasize that none of our modeling results hinge on this decision. We have made two changes to Supplementary Figure 6 to address these points: To address the latter point, we now exclude the “7” and “8” rows from the plots in the figure (since the model did not have to predict another state after states 7 and 8). To address the former point (about the 1-2 to 3-4 transition not being shown to the model), we have amended the figure caption as follows (new text in **bold**):

Analysis of the number of inferred latent causes and their properties. (a) Histogram of the number of inferred latent causes in each of the model conditions for the model used in Simulation 3. (b) Plot of the learned transition matrices in Simulation 3 in the case where the model infers two latent causes (in either the interleaved or blocked condition), and (c) plot of the learned transition matrix in Simulation 3 in the case where the model only infers one latent cause (in the interleaved condition). When the model infers two latent causes, the learned transition matrices accurately reflect the ground truth transition structure; when the model only infers one latent cause, it learns a transition matrix that averages across the ground-truth transition matrices for the two Markov chains. **Note that the rows corresponding to the transition from states 1 and 2 to the following state are empty because this unpredictable transition was not shown to this version of the model (see discussion of this point in Simulation 2, and also in the Limitations section of the Discussion).**

Lastly, we have added the colorbar requested by the reviewer.